# The cryo-EM structure of the acid activatable pore-forming immune effector Macrophage-expressed gene 1

Siew Siew Pang[1,2,13], Charles Bayly-Jones[1,2,13], Mazdak Radjainia[2,3], Bradley A. Spicer[1,2], Ruby H.P. Law [1,2], Adrian W. Hodel[4,5], Edward S. Parsons[4], Susan M. Ekkel[1,2], Paul J. Conroy [1,2], Georg Ramm [2], Hariprasad Venugopal[2], Phillip I. Bird [2], Bart W. Hoogenboom [4,5,6], Ilia Voskoboinik[7,8], Yann Gambin[9,10], Emma Sierecki[9,10], Michelle A. Dunstone[1,2] & James C. Whisstock[1,2,11,12]

Macrophage-expressed gene 1 (MPEG1/Perforin-2) is a perforin-like protein that functions within the phagolysosome to damage engulfed microbes. MPEG1 is thought to form pores in target membranes, however, its mode of action remains unknown. We use cryo-Electron Microscopy (cryo-EM) to determine the 2.4 Å structure of a hexadecameric assembly of MPEG1 that displays the expected features of a soluble prepore complex. We further discover that MPEG1 prepore-like assemblies can be induced to perforate membranes through acidification, such as would occur within maturing phagolysosomes. We next solve the 3.6 Å cryo-EM structure of MPEG1 in complex with liposomes. These data reveal that a multivesicular body of 12 kDa (MVB12)-associated β-prism (MABP) domain binds membranes such that the pore-forming machinery of MPEG1 is oriented away from the bound membrane. This unexpected mechanism of membrane interaction suggests that MPEG1 remains bound to the phagolysosome membrane while simultaneously forming pores in engulfed bacterial targets.

[1] ARC Centre of Excellence in Advanced Molecular Imaging, Monash University, Melbourne, VIC 3800, Australia. [2] Biomedicine Discovery Institute, Department of Biochemistry and Molecular Biology, Monash University, Melbourne, VIC 3800, Australia. [3] Thermo Fischer Scientific, Achtseweg Noord 5, Building 5651 GG, Eindhoven, The Netherlands. [4] London Centre for Nanotechnology, University College London, London WC1H 0AH, UK. [5] Institute of Structural and Molecular Biology, University College London, London WC1E 6BT, UK. [6] Department of Physics and Astronomy, University College London, London WC1E 6BT, UK. [7] Cancer Immunology Program, Peter MacCallum Cancer Centre, Melbourne, VIC 3000, Australia. [8] Department of Genetics, The University of Melbourne, Parkville, VIC 3010, Australia. [9] School of Medical Science, University of New South Wales, Randwick, NSW 2052, Australia. [10] EMBL Australia, Single Molecule Node, University of New South Wales, Sydney 2052, Australia. [11] EMBL Australia, Monash University, Melbourne, VIC 3800, Australia. [12] ACRF Department of Cancer Biology and Therapeutics, John Curtin School of Medical Research, Australian National University, Canberra, ACT 2601, Australia. [13] These authors contributed equally: Siew Siew Pang, Charles Bayly-Jones. Correspondence and requests for materials should be addressed to M.A.D. (email: Michelle.Dunstone@monash.edu) or to J.C.W. (email: James.Whisstock@monash.edu)

Macrophage-expressed gene 1 (MPEG1, also termed Perforin-2) is one of the most ancient and highly conserved members of the membrane attack complex (MAC)/perforin-like (PF)/cholesterol-dependent cytolysin (MACPF/CDC) superfamily, with orthologous sequences identifiable throughout the metazoan[1–4]. MACPF/CDC superfamily members play diverse roles in immunity, neural development, insect development and bacterial pathogenesis[5]. In the context of the perforin branch of the superfamily, phylogenetic studies suggest that MPEG1 represents the archetype of the human immune pore-forming MAC and perforin[6].

The majority of MACPF/CDC proteins are produced as soluble monomers that bind to membranes via ancillary lipid or protein receptor-binding domains and that then self-assemble into arc- or ring-shaped pores (Supplementary Fig. 1)[7–11]. For many MACPF/CDCs, pore formation proceeds via assembly of a transient, intermediate prepore state, in which the membrane spanning regions have not yet been released[12–15]. A conformational change within each MACPF/CDC domain in the prepore assembly then permits two small clusters of α-helices (termed transmembrane hairpin (TMH)-1 and -2) to unravel and penetrate the membrane as two amphipathic β-hairpins (Supplementary Fig. 1)[9,16]. The final pore comprises a large (complete or incomplete) β-barrel that spans the membrane[14,17].

MPEG1 traffics throughout the endosomal pathway, ultimately localising to the late-endosome and phagosome[18,19]. MPEG1 expression is reported to be constitutively driven in phagocytes[20]. However, pro-inflammatory molecules, such as TNFα and LPS, have also been shown to induce expression in parenchymal cells[19,21]. Moreover, these triggers induce ubiquitination of the MPEG1 cytosolic region, resulting in redistribution and trafficking of MPEG1 to the late-endosome and phagolysosome[19,22].

In contrast to other perforin-like proteins characterised to date, human MPEG1 is an integral type-1 transmembrane protein. The mature protein is comprised of three regions; an ectodomain located in the vesicular lumen, a transmembrane domain anchor that spans the vesicular membrane and, finally, a short cytosolic sequence (Supplementary Fig. 2). The 636 amino-acid ectodomain consists of an N-terminal MACPF/CDC domain together with a C-terminal region that is uncharacterised in terms of structure and function[18,20,23] (Supplementary Fig. 2).

Published studies suggest that MPEG1 functions within the macrophage phagolysosome as a pore-forming protein[18,19,24]. Indeed, although a bactericidal pore-forming function for MPEG1 has not yet been definitively identified in vivo or in vitro, bacterial membranes derived from macrophages are decorated with ring-like structures, suggesting that MPEG1 is an immune effector[18]. In these regards, some patients suffering from pulmonary nontuberculous mycobacterial infections carry MPEG1 mutations[25].

As such, MPEG1 appears to have an immunological function. In order to better understand the nature of this role, we express and purify the human form of MPEG1 and find MPEG1 membranolytic activity can be induced upon acidification. We further determine the 2.4 Å and 3.6 Å cryo-EM structures of MPEG1 in a soluble prepore-like assembly and bound to liposomes, respectively. Remarkably, these data show MPEG1 binds to membranes inversely compared to other MACPF members, suggesting MPEG1 can form pores in a second, opposing membrane while remaining tethered to the inner leaflet of the phagolysosome membrane.

## Results

### The cryo-EM structure of the soluble MPEG1 prepore. Previous work suggests that MPEG1 ectodomain may be proteolytically

cleaved away from the transmembrane domain[18]. This region can also be expressed as a splice variant lacking the transmembrane/cytosolic region[19]. We therefore expressed the human MPEG1 ectodomain in order to investigate its function. Electron microscopy experiments revealed that the majority of recombinant material eluted as a stable multimeric complex (Fig. 1a). Accordingly, an integrative single particle cryo-EM approach was used to elucidate the structure of this assembly (Fig. 1b). Our initial efforts yielded a 3.5 Å structure of wild-type MPEG1. These data revealed that MPEG1 formed hexadecameric rings stacked together to form a double ring (or head-to-head assembly). Although this structure was incomplete, particularly in the C-terminal region, these data provided a platform for directed mutagenesis. Indeed, our (ultimately unsuccessful) attempts to disrupt the interface between the double ring serendipitously led to our identifying the MPEG1 L425K mutation as yielding a greatly improved sample and an improvement in global resolution to 2.4 Å (workflow described in detail in the methods). This mutation resulted in two different MPEG1$_{L425K}$ assemblies, one of which involved domain swapping between the two head-to-head coordinated rings. Ultimately, a cohort of structures was determined, modelled and interpreted (Fig. 1b, Supplementary Fig. 3).

The final 2.4 Å structure of the MPEG1 assembly revealed two hexadecameric rings stacked together in the aforementioned head-to-head arrangement. Each monomer of the 16-subunit ring possesses an N-terminal MACPF/CDC domain, a central multivesicular body of 12 kDa (MVB12)-associated β-prism (MABP) domain and a C-terminal region (termed the L-domain) (Fig. 1c, Supplementary Fig. 4). The structure confirmed previous bioinformatic prediction of the MACPF/CDC domain[6,20]. In contrast, the MPEG1 MABP domain possesses <5% primary amino-acid sequence similarity to other homologues and the fold was accordingly identified using Dali searches[26]. The L-domain directly precedes the transmembrane region and includes a short two-stranded β-sheet capped by an α-helix.

In contrast to other MACPF/CDCs studied to date[12,27–30], the recombinant form of MPEG1 has pre-assembled into a soluble oligomer that exhibits the expected features of a prepore assembly. Within each subunit, both TMH-1 and -2 are folded as small helical bundles (Fig. 1c). A central feature of the assembly is a 64-stranded β-barrel that represents the top, preformed portion of the future pore. The majority of MACPF/CDC domain-mediated inter-subunit interactions are formed around the central lumen of the β-barrel (Fig. 1d, Supplementary Table 1). Around the outside of the assembly both the MABP domain and the L-domain also mediate extensive inter-subunit interactions (Fig. 1e). The L-domain forms substantial inter-subunit interactions in trans with an extraordinary elongated β-hairpin that extends from the MABP domain of the adjacent subunit (the MABP β-hairpin; Fig. 1c, e; Supplementary Table 1). Finally, we note that one of the mutations associated with human disease (T73A[25]; corresponding to T56 in our structure) maps to a loop buried within the inter-subunit interface. Changes in this region, thus, may lead to oligomer instability or reduced capacity to function (Fig. 1d).

Three-dimensional classification experiments and multibody analysis of the head-to-head assembly reveal that one ring can adopt multiple positions relative to the other ring, that is, neither ring possesses a single well-defined orientation (Supplementary Figs. 3, 5). Similar double-ring structures have been observed in other pore-forming proteins (e.g. aerolysin[31]; gasdermin[32]) and can form as a consequence of two membrane-binding surfaces interacting with each other. Indeed, in these regards the MPEG1 interaction is mediated by the interface defined by the MABP. This is likely to be an in vitro effect of our recombinant construct, which lacks the transmembrane region (Supplementary Figs. 2, 4).

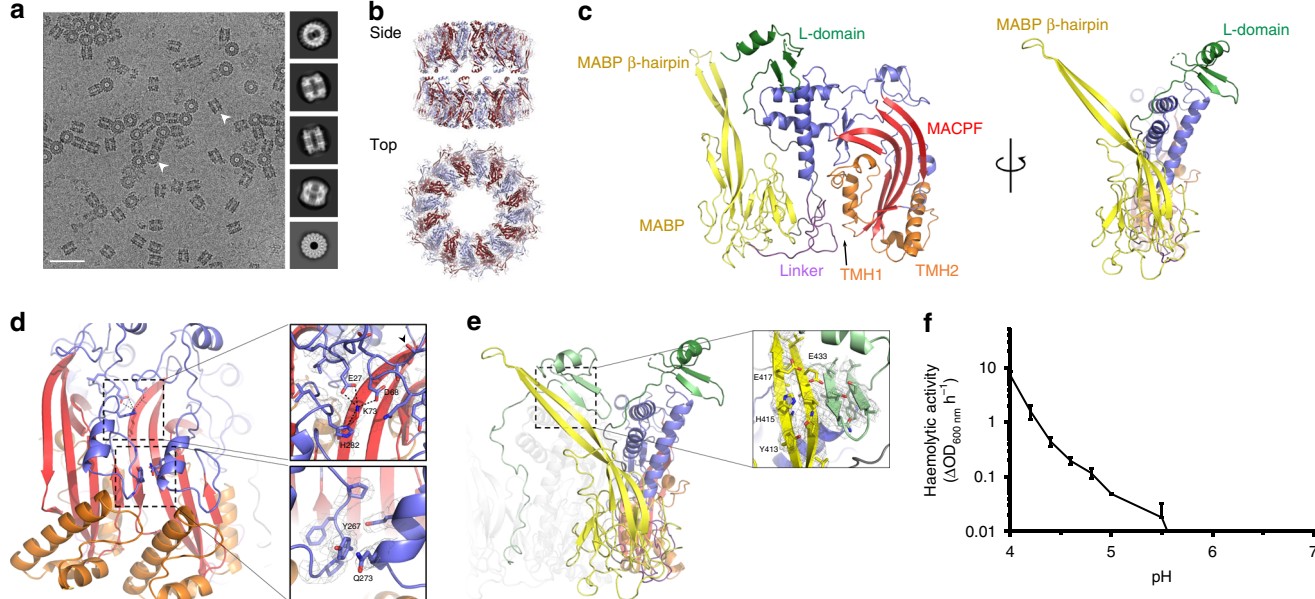

**Fig. 1** Cryo-EM of MPEG1 soluble prepores. **a** Representative image of MPEG1 together with a selection of class averages are shown (scale bar: 50 nm). **b** 2.4 Å structure of the MPEG1 assembly; two hexadecameric rings (monomers alternately coloured) stack together to form a head-to-head dimer. **c** Two views of the MPEG1 monomer are shown (side on (left) and the C-terminal region (foreground) viewed from peripheral region of assembly (right)). The MACPF/CDC domain is in blue, with the central sheet in red, and TMH-1/2 in orange. The central globular domain (MABP domain) is in yellow, and the L-domain is in green. **d** Interactions made in the lumen, around the central β-sheet. Residue T56 is indicated by an arrow. **e** Interactions made in trans by the MABP β-hairpin (yellow) with the L-domain (green). **f** Lytic activity of the prepore material at a range of different pH (min. to max. box limits, error bars represent SE, $n = 4$. Error is plotted for all points). Source data are provided as a Source Data file

**Acid-induced membranolytic activity**. We next investigated whether the MPEG1 prepore could be induced to form pores. At neutral pH, the recombinant material possessed no detectable activity in red blood cell (RBC) lysis assays (Fig. 1f). However, we found that MPEG1 started to gain membranolytic activity at acidic pH (<5.5) (Fig. 1f). To date, only one other acid activated MACPF/CDC protein has been identified, which is the bacterial cytolysin, listeriolysin O (LLO)[33,34]. The latter protein possesses enhanced pore-forming activity at low pH and functions to facilitate pathogen survival via damaging the phagolysosome[35]. Furthermore, the subcellular localisation of MPEG1 within the endosomal pathway is consistent with MPEG1 encountering pathogens in a low pH environment.

As MPEG1 is not active at neutral pH in vitro, our data suggest that pH may be a regulatory mechanism preventing activation until MPEG1 reaches the terminal compartments of the endosomal pathway. This mechanism would help guard against premature off-target membranolytic activity. In support of this idea, previous studies have found that MPEG1 activity is dependent on transition to these acidic compartments[22]. Indeed, abolition of MPEG1 ubiquitination stalls subsequent trafficking from the endosome to later more-acidic compartments, causing endosomal vesicles to become enriched with MPEG1. These cells, resultantly, lack MPEG1 function, indicating that MPEG1 activity is spatially regulated and dependent on trafficking to these acidic compartments[22].

**Interaction of MPEG1 prepores with lipid bilayers**. We reasoned that the MABP domain, a fold known to bind lipids[36], likely functions to localise MPEG1 to target membranes. Previous studies reveal that MABP proteins interact with acidic membranes via a positively charged loop[36]. Superposition of the MABP domain of MPEG1 with the archetypal homologue from the endosomal sorting complexes required for transport-I

MVB12 subunit[36] revealed, however, that the membrane-binding loop maps to the extended MABP β-hairpin (Fig. 2a). These data, thus, suggested that the MABP domain is inverted with respect to the membrane-binding domains of conventional MACPF/CDC proteins such as perforin and LLO (Supplementary Figs. 1, 4).

Given these findings, we sought to understand MPEG1 lipid specificity and how MPEG1 prepores coordinate membranes. Initial lipid screening by membrane lipid and PIP strips suggested that MPEG1, like the MVB12 MABP domain, displays a preference for negatively charged lipids such as POPS, PIPs and cardiolipin (Fig. 2b). Negatively charged lipids were hence empirically chosen to yield data suitable for cryo-EM. We next determined the 3.6 Å cryo-EM structure of the MPEG1 prepore assembly bound to POPC/POPS liposomes at neutral pH (Fig. 2c). These data revealed the MPEG1 MABP binds lipids in a canonical fashion through the MABP β-hairpin. The latter region bends as a consequence of interactions with the membrane, a shift that involves breaking the in trans contacts with the L-domain of the adjacent subunit (Fig. 2d). Indeed, the C-terminal portion of the L-domain is not visible in the electron density, suggesting that this region becomes flexible when the MABP β-hairpin distorts through membrane interactions (Fig. 2d). The remainder of the structure appears essentially unaltered (Fig. 2c, d). We also investigated MPEG1 binding to liposomes derived from *Escherichia coli* membranes, as potential targets for MPEG1 pore formation; these data reveal the same orientation of MPEG1 prepore binding (Supplementary Fig. 6).

Atomic force microscopy (AFM) studies confirm the binding of MPEG1 prepores to both POPC/POPS (Supplementary Fig. 7) and *E. coli* lipid membranes (Fig. 3). Consistent with previous studies on perforin and CDC prepores on supported lipid bilayers[9,10], MPEG1 prepores appear as mobile (and therefore poorly resolved) features on the membrane (Fig. 3a). Upon

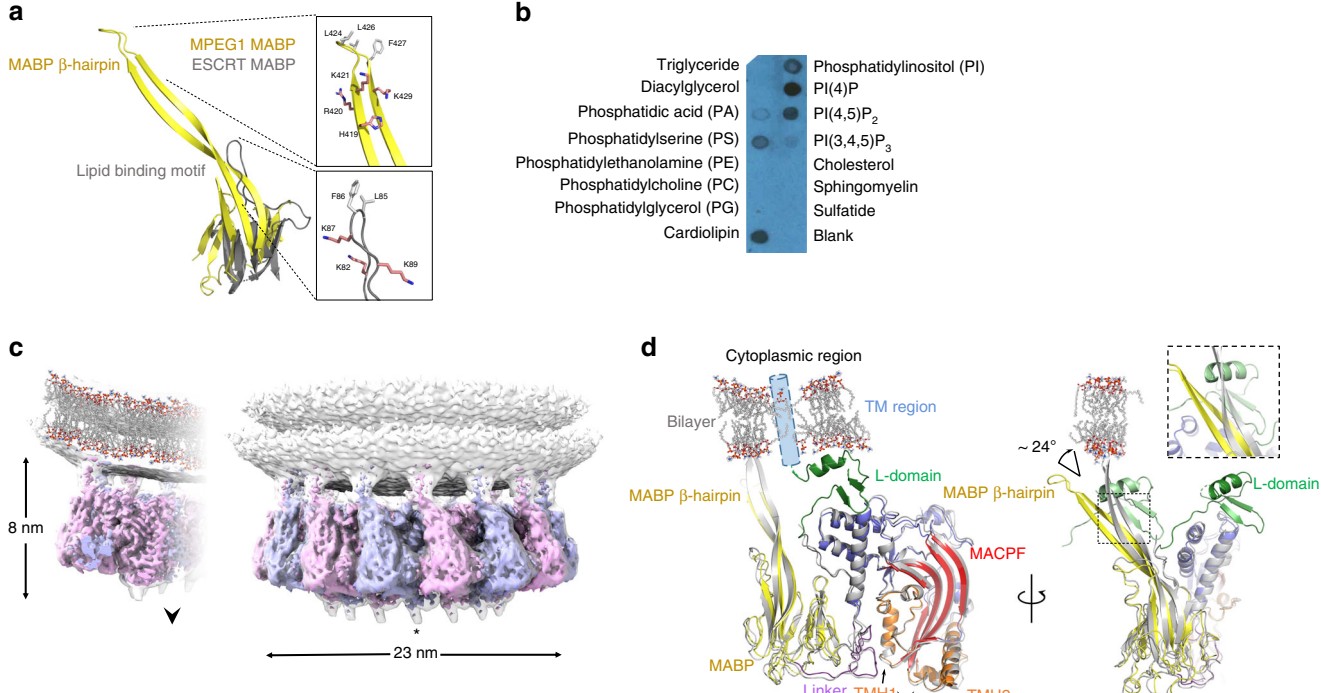

**Fig. 2** MPEG1 interacts with negatively charged phospholipids via its MABP β-hairpin. **a** Structural superposition of the MPEG1 MABP domain (yellow) and the MVB12-associated β-prism (grey). The lipid-binding loop of the MVB12-associated β-prism maps to the MABP β-hairpin. The tip of both the MPEG1 β-hairpin and the MVB12 loop similarly contain a group of positively charged and hydrophobic residues. **b** MPEG1 displays broad specificity for lipids with negative charge. **c** 3.6 Å structure of MPEG1 bound to lipid membranes (subunits alternately coloured). A glycan moiety (indicated by an asterisk) is attached to TMH-2. The direction that the pore-forming β-hairpins are released to form a membrane spanning β-barrel is shown by an arrow. **d** Structural superposition of the MPEG1 monomer derived from the head-to-head assembly (coloured as in Fig. 1c), and the lipid bound form (grey). The predicted position of the transmembrane domain (absent in the structure) is shown (blue cylinder). The β-hairpin shifts ~24° in response to lipid interaction and the L-domain is disordered in the lipid bound form. The approximate position of the membrane is shown for reference

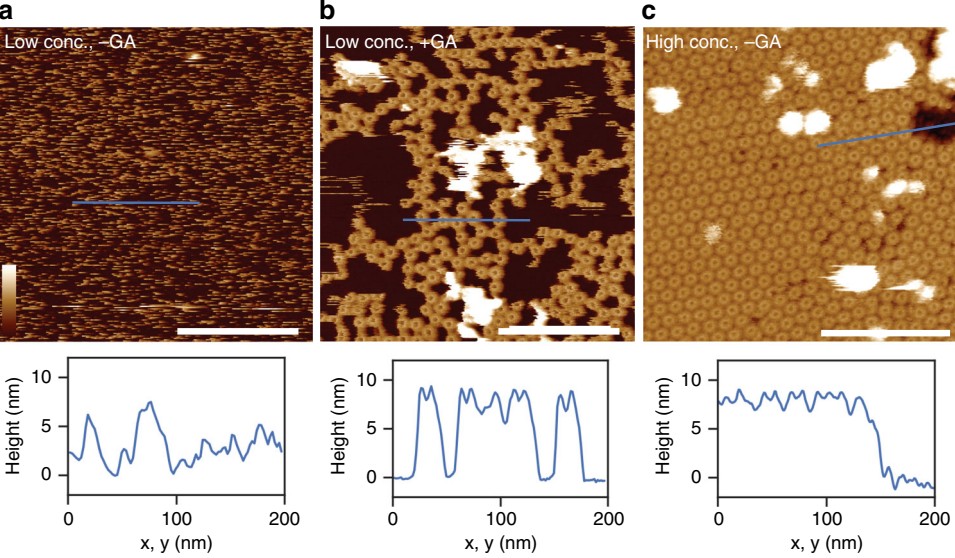

**Fig. 3** Atomic force microscopy images of MPEG1 on supported lipid bilayers consisting of *E. coli* total lipid extract. **a** Without (-GA) and **b** with (+ GA) glutaraldehyde fixation, at neutral pH. **c** Increasing the concentration of MPEG1 results in a high-density, hexagonal packing on the supported bilayer. The dynamic motion observed in **a** is therefore reduced, such that the MPEG1 assemblies could be well resolved without glutaraldehyde fixation. Shown below each AFM images are 1D height profiles extracted from selected regions (blue lines) where both membrane and MPEG1 prepores are observed. When packed in dense hexagonal lattice, the MPEG1 assemblies mostly occlude the membrane from the AFM probe, resulting in a smaller height variation (until a region of bare membrane is encountered). Scale bar: 200 nm. Colour (height) scale: 16 nm

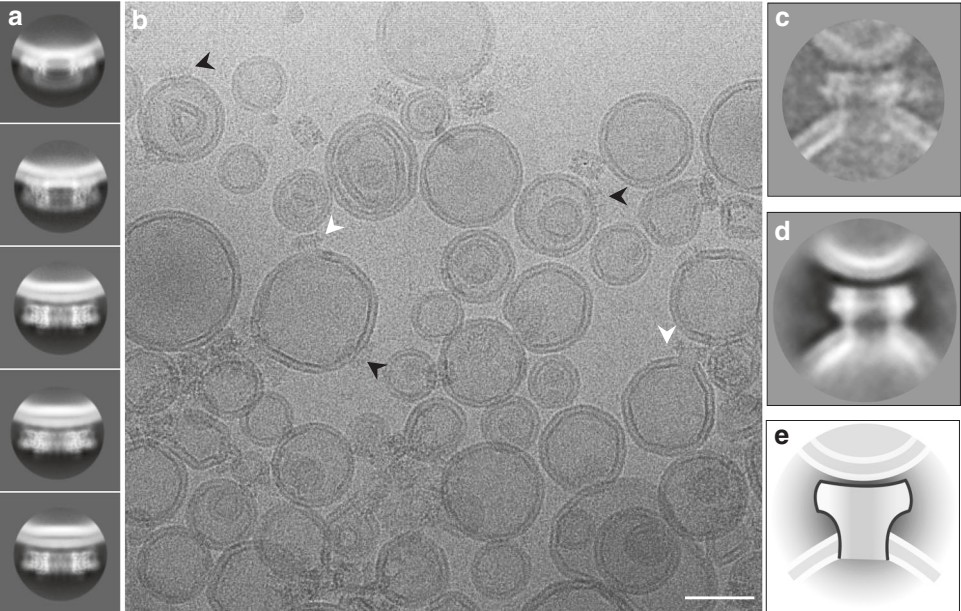

**Fig. 4** The MACPF domain of MPEG1 is oriented away from the MABP-bound membrane. **a** Examples of MPEG1 bound to liposomes reveal **b** single rings bound in the prepore state (black arrows) and the occasional example of pores (i.e., the membrane is absent in the pore lumen), that coordinate a second liposome through the top portion of the ring (white arrow). We were not able to identify examples of pores bound to a single liposome, i.e. all pore structures we observed bridged between two liposomes. Class averages for prepores (**a**) and pores (**d**) are shown. For comparison, **c** a single pore, **d** a 2D class average and **e** illustration are shown

fixation with glutaraldehyde, these features can be clearly identified as ring-shaped assemblies of ~12–13 nm diameter (prepore rim, peak-to-peak), fully consistent with the electron microscope (EM) data (Fig. 3b). The height of the prepores is ~8 nm above the membrane surface, which agrees with the height of a single-ring MPEG1 prepore as estimated based on the EM data (Fig. 2). Increasing the concentration of protein forced MPEG1 prepore assemblies to pack into a hexagonal lattice, reducing their mobility and, therefore, enabling us to resolve MPEG1 without glutaraldehyde fixation (Fig. 3c, Supplementary Fig. 7c). Collectively, these data suggest that, like the MVB12 MABP domain, the MPEG1 MABP domain has relatively broad specificity for a range of different lipids.

The observed mode of MPEG1 membrane binding is also consistent with the location of the C-terminal transmembrane region (absent in our construct; Fig. 2d, Supplementary Fig. 2), which, in full-length protein, immediately follows the L-domain. Taken together, these data suggest that MPEG1 includes two independent features—a transmembrane sequence and the MABP β-hairpin—that could each function redundantly to ensure localisation of the protein to the inner leaflet of the vesicular membrane. Such an interaction mode, if it were to be maintained during lytic function, would mean that the pore-forming regions released by the MACPF domain would point directly away from membrane coordinated by the MABP domain (Fig. 2c, d). The implication of our findings is therefore that MPEG1 may be able to bind one membrane system through the MABP domain while simultaneously forming pores in a second membrane system. In support of this idea, analysis of the liposome data set collected at neutral pH revealed occasional examples of MPEG1 apparently coordinated to one liposome, while forming pores in a second (Fig. 4). However, there were only a few examples of such complexes present (typically 1–2 pores/image) and it was not possible to collect a data set. Further, despite repeated efforts, incubation of MPEG1/liposome mixtures at acidic pH failed to yield a sample suitable for collection of a

high-quality data set due to extensive protein and liposome aggregation.

## Discussion
In this study, we expressed and structurally characterised the ectodomain of human phagolysosomal protein MPEG1. Our data revealed a number of unexpected findings. In contrast to other MACPF/CDC proteins characterised to date, the MPEG1 ecto-domain can preassemble into a prepore form. Further, bio-chemical studies suggest that this assembly can be triggered to form pores in vitro upon exposure to acidic pH. This feature of the molecule may help prevent unwanted or unproductive pore formation at neutral pH, and lends support to the idea that MPEG1 functions to form pores in bacteria located within late acidified phagolysosomal compartments[18]. Most remarkably, however, our structural data, together with the location of the type-I transmembrane domain in full-length protein, suggests that the lipid-binding MABP domain of MPEG1 may function to bind to the inner leaflet of the phagolysosomal membrane. Such an interaction mode would be anticipated to protect against MACPF pore formation in the bound host membrane (Fig. 2c, d). This contrasts with other MACPF proteins, such as perforin and the CDCs, which deploy ancillary lipid-binding domains to directly assemble on the membrane targeted for pore formation (Supplementary Figs. 1, 4).

Given these findings, we speculated how MPEG1 function might form pores in engulfed targets. One possibility is that other intra-cellular events, for example, proteolysis[18] prior to pH change, permits release of a portion of MPEG1 into the lumen of the phagosome where it may function to form pores in a conventional perforin/CDC-like fashion (Fig. 5a). Alter-natively, we suggest that the observed mode of MABP mem-brane binding may permit interaction with the phagolysosome membrane while simultaneously forming pores in engulfed bacteria (Fig. 5b). Indeed, such a mechanism may be

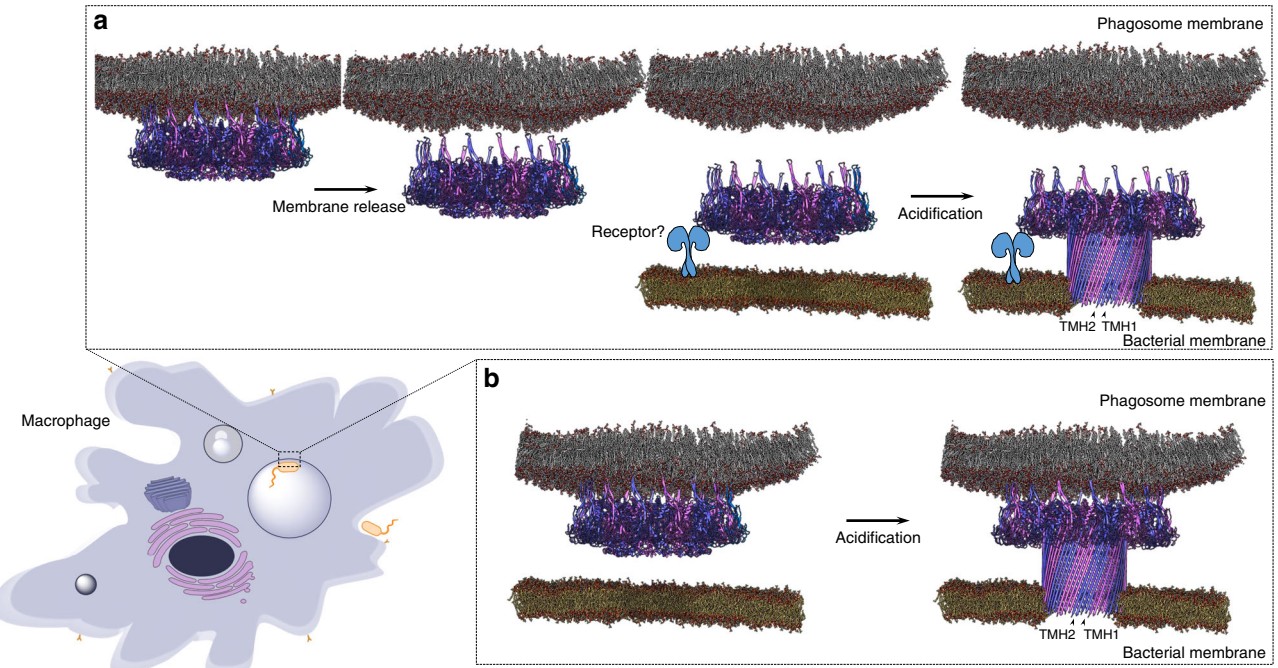

**Fig. 5** Schematic illustrating two proposed mechanisms for control of MPEG1 function. **a** MPEG1 prepores (alternate subunits coloured) are localised to the phagolysosome membrane, either through the transmembrane helix or the MABP β-hairpin. Upon acidification or proteolytic activity, the prepores are released from the membrane into the phagosome lumen. Via an unknown means these recognise engulfed target membranes and form pores. **b** Alternatively, prepores remain tethered to the phagosome membrane inner leaflet via the MABP β-hairpin and are triggered to form pores in engulfed bacteria upon acidification

advantageous in regards to preventing unwanted autolysis of the phagolysosome.

## Methods

**Protein purification.** The human *MPEG1* gene for the soluble ectodomain (NCBI-gene ID: 219972, nucleotide 49–1959) was synthesised (Genescript) with codon usage optimised for insect cell expression. The native signal peptide was replaced with the honeybee melittin signal peptide for protein secretion, and a C-terminal hexahistidine tag was introduced for affinity purification. Unique restriction sites *Eco*RI and *Xho*I were also introduced at the 5′- and 3′-ends so that the recombinant gene could be sub-cloned into pFastBac1 expression vector (Life Technologies) for bacmid production. Baculoviral stocks (P1 to P3) were generated in Sf9 (Thermo Fisher Scientific, #11496015) or Sf21 insect cells (Thermo Fisher Scientific, #11497013) as described in the supplier's protocols.

For MPEG1 expression, Sf21 cells ($2 \times 10^6$ cells mL$^{-1}$) were infected with the P3 viral stock in Insect-XPRESS protein-free medium (Lonza). The infected insect cells were grown at 27 °C with shaking for 66 h. The insect cell supernatant was harvested by centrifugation at $1000 \times g$ for 10 min. The clarified supernatant was cooled to 4 °C and buffer-exchanged into 20 mM Tris-HCl, pH 8.0, 0.3 M NaCl, 20 mM imidazole by extensive dialysis or by tangential flow filtration (Cogent M1 TFF system, Millipore).

The buffer-exchanged supernatant was clarified by filtering through a 0.8 μm membrane before loading onto a Ni-NTA agarose column (Qiagen). The column was washed with buffer containing 40 mM imidazole and the His-tagged MPEG1 eluted with buffer containing 500 mM imidazole. The protein peak fractions were pooled and dialysed against 20 mM Tri-HCl, pH 7.2, 0.3 M NaCl, 10 % (w/v) glycerol at 4 °C overnight. The pooled protein was further purified by size-exclusion chromatography on a Superose 6 10/300 column (GE Healthcare Life Sciences). MPEG1 fractions were monitored using sodium dodecyl sulfate polyacrylamide gel electrophoresis (SDS-PAGE), haemolytic activity assay and negative-stained EM. For further cryo-EM work, MPEG1 fractions were selected and pooled carefully based on negative-stained EM analysis for sample homogeneity. For all other work, fractions were assessed and pooled based on purity on SDS-PAGE and haemolytic activity (Supplementary Fig. 8a). Pooled fractions were concentrated to 0.5–1.0 mg mL$^{-1}$ and snap-frozen in liquid nitrogen for storage at −80 °C.

A similar approach was used to produce a L425K mutant form of MPEG1 (Supplementary Table 2). The information derived from the MPEG1$_{L425K}$ proved important for determining the complete structure of wild-type protein and is described in the model building and analysis section.

**Lipid-binding screen.** The membrane lipid strips were purchased from Echelon Biosciences. The lipid-binding screen was carried out as recommended by the supplier with some modifications. The strips were blocked with 1 % (w/v) skim milk in Tris-buffered saline (TBS) (10 mM Tris-HCl, pH 8.0, 150 mM NaCl) overnight at 4 °C. The blocked strips were incubated with 10 μg mL$^{-1}$ MPEG1 in TBS-T (TBS and 0.1 % (v/v) Tween 20) at room temperature for 1 h. The strips were washed three times in TBS-T before probing with anti-6 × His tag-horseradish peroxidase-conjugated antibody (Abcam) to detect lipid-protein interactions.

**RBC lysis assay.** Rabbit blood was obtained from Applied Biological Products Management and collected in the presence of heparin to prevent clotting. To prepare the RBCs for the lysis assays, the blood was first fractionated by centrifugation at $3000 \times g$ at 4 °C for 15 min to pellet the RBC. The cells were resuspended gently and washed three times with equal volume of HEPES buffered saline (HBS). The final RBC pellet was resuspended in preservative Celpresol (CSL) to the original blood volume for storage at 4 °C.

Before each lysis assay, the rabbit RBC were pelleted and washed in wash buffer (5 mM HEPES pH 7.0, 75 mM NaCl, 2.5 % (w/v) glucose, 0.15 mM CaCl$_2$, 0.5 mM MgCl$_2$) three times to remove the preservative solution, any lysed cells and soluble haemoglobin. MPEG1 membranolytic activity was monitored by reduction of turbidity of RBC suspension at OD$_{600nm}$. The washed RBC were diluted 20-fold into reaction buffer (20 mM buffer, 75 mM NaCl, 2.5 % (w/v) glucose) with the respective pH (sodium acetate buffer for pH 4.5–5.5, MES buffer for pH 6.0–6.5 and Tris-HCl for pH 7.0) to give a starting OD$_{600nm}$ ~1.5–2.0. The haemolytic activity was initiated by adding the diluted RBC to MPEGl. MPEG1 membranolytic activity was calculated by the rate of RBC lysis (ΔOD$_{600nm}$ h$^{-1}$) per μg of MPEG1 protein. All reactions were carried out in duplicate and blanked against identical assays except without the addition of MPEG1, four independent experiments were performed ($n = 4$) (Supplementary Fig. 8b).

**Cryo-EM sample preparation and data collection.** For cryo-EM of MPEG1, the protein sample was buffer-exchanged into 20 mM Tris-HCl, pH 7.2, 0.3 M NaCl to remove the glycerol, and concentrated between 2.0 and 2.5 mg mL$^{-1}$. The cryo-EM grids were generated using the Vitrobot System (Thermo Fisher Scientific). Initial grid freezing conditions were tested and screened on a Tecnai T12 electron microscope (Thermo Fisher Scientific). In brief, 3 μL of MPEG1 was applied to a glow-discharged QUANTIFOIL Cu R 1.2/1.3 grid, with blotting conditions as follows. The temperature was set to 4 °C with the relative humidity option turned off, the grids were blotted with a blot time of 2.5 s, blot force of −1 and drain time of 1 s. The grids were snap-frozen in liquid ethane and stored under liquid nitrogen until TEM data collection.

For cryo-EM of liposome/MPEG1 complex, the protein solution was prepared as described above into (20 mM HEPES pH 7.0, 0.15 M NaCl). Two types of lipid compositions were used; liposomes of POPC and POPS (Avanti Polar Lipids) in equal ratio or liposomes of *E. coli* total lipid extract (ETL, Avanti Polar Lipids). Both liposomes were made in a similar fashion. To prepare the liposomes, 4 mg of chloroform-solubilised lipid (2 mg of each POPC and POPS or 4 mg of ETL) was dried under argon in a clean test tube, and desiccated under vacuum for at least 4 h. The dried lipid mixture was resuspended in 0.5 mL HBS buffer by vortexing for 1 min, then snap-frozen in liquid nitrogen, and sonicated in a warmed (30 °C) ultrasonic bath for 15 min. This process was repeated three times. The lipid suspension was then extruded using a polycarbonate membrane with pore size of 0.1 μm (Avanti Polar Lipid) to obtain unilamellar liposomes. To generate liposome/ MPEG1 complex, 10 μL POPC:POPS liposomes was mixed with 5 μL MPEG1 (1.7 mg mL$^{-1}$) and incubated at 37 °C for three hours. The liposome and MPEG1 mixture was frozen onto a glow-discharged QUANTIFOIL Cu R 2/2 grid as described above with the following modifications. The Vitrobot conditions were set to 22 °C with 100 % humidity, the grids were blotted with a blot time of 2.5 s, blot force of −5 and drain time of 1 s.

Data collection parameters have been summarised (Supplementary Table 3) for each data set. In brief, dose-fractionated movies were collected on a Titan Krios (Thermo Fisher Scientific), equipped with a Quantum energy filter (Gatan) and Summit K2 (Gatan) or a Falcon II (Thermo Fisher Scientific) direct electron detector. Data acquisition was performed using either SerialEM[37] or EPU (Thermo Fisher Scientific).

**Cryo-EM image processing**. Upon finalisation of data collection to calibrate pixel size and estimate magnification anisotropy, images of gold diffraction grating (Agar Scientific) were collected at the same magnification as the collection. Analysis performed with mag_distortion_estimate[38] indicated magnification anisotropy was indeed present (Supplementary Table 3). Therefore, dose-fractionated movies were corrected for beam induced motion, anisotropy and radiation damage within MotionCor2[39]. Super-resolution movies were additionally down sampled by a factor of 2, applied by Fourier cropping within MotionCor2. All aligned movie frames were subsequently averaged into dose-weighted and non-weighted sums for further processing.

Particle coordinates were determined using various software depending on the data set, a combination of Gautomatch (Zhang et al., unpublished; https://www.mrc-lmb.cam.ac.uk/kzhang/Gautomatch/), crYOLO (v1.1)[40] and manual picking were employed. A rapid single round of 2D classification in cryoSPARC[41] (v1) was employed to remove contaminants and false positives. Contrast transfer function (CTF) estimation of whole non-dose-weighted micrographs was initially performed with CTFFIND4 (v4.1.10)[42]. On-the-fly processing was performed in RELION (v2.1)[43] to assess data quality. Initial models were all generated ab initio in either cryoSPARC or in EMAN[44] (v2.2) by the common line method for the MPEG1$_{WT}$ data set (Supplementary Fig. 3). Symmetry was determined from 2D and 3D class averages and correct symmetry was confirmed by comparison with C1 reconstructions. Ultimately different symmetries were imposed depending on the reconstruction ranging from full D16, C16 or C1.

All further processing was performed in RELION (v1.4, v2.1 and v3.0), unless otherwise stated. Particles belonging to clean classes were then subjected to 3D classification to remove malformed particles and projections with poor signal-to-noise. Clearly abnormal classes were discarded, and 3D refinement was performed on the remainder. Per-particle CTF parameters were refined without further alignments in cisTEM (v1.0b)[45] until convergence, here the refinement resolution was maintained above the fall off of the FSC. Particles with poor scores from cisTEM were discarded.

Masked 3D classification with the previously refined angular and CTF parameters without additional alignments was carried out for all data sets to identify homogeneous subpopulations. In the case of the soluble MPEG1$_{L425K}$ data set, this resulted in two distinct populations. One population exhibited a discrete D16 conformational state, termed the β-conformation (MPEG1$_{L425K}$β), and consisted of approximately one-sixth of all the particles (Supplementary Figs. 3, 9). This conformer displayed isotropic resolution and enabled the MABP domain to be resolved. The second population, termed the α-conformation (MPEG1$_{L425K}$α), exhibited continuous conformational heterogeneity. This population could be further reduced into a continuous distribution of D16 states. Here conventional refinement failed, resulting in reconstructions that had radially decreasing quality where only the central most region was resolved. Analysis by RELION (v3.0b2) multibody procedure[46] resulted in a marked improvement in resolution. Indeed, principal component analysis highlighted additional degrees of freedom and loose association of each ring (Supplementary Fig. 3). Symmetry breaking hence occurred as a result of relative motion between each ring.

To overcome symmetry breaking, localised reconstructions were performed on each ring and additionally on individual monomers, thereby reducing the symmetry to C16 and C1, respectively. First, partial signal subtraction of both the top and bottom rings was performed in RELION to obtain two sub-particles per image. These were individually refined in cryoSPARC with dynamic masking. Localised reconstruction resulted in a notable improvement in quality of the reconstruction and corresponding FSC. Resultantly, a similar analysis was performed on the MPEG1$_{WT}$ data set, also yielding an improvement from 3.5 to

2.9 Å (although the advent of newer software from RELION-1.4 to 3.0 may also explain this improvement). Later sub-particles were merged into a single data set, which refined to the highest resolution in the central region of the complex (MACPF domain) (Supplementary Fig. 10a). However, owing to intramolecular, conformational heterogeneity between subunits, the quality of the reconstruction in the MABP domain remained poor and hence the electron density was uninterpretable (Supplementary Fig. 10b). Therefore, sub-particles corresponding to individual monomers were re-extracted along with partial signal subtraction using the localised reconstruction scripts[47]. These sub-particles were aligned to a common axis and sorted by masked 3D classification without further alignments, where a subpopulation with substantially higher homogeneity was identified thereby enabling the full structure to be resolved (Supplementary Fig. 10c, d).

In the case of the liposome/MPEG1 complex, density subtraction of the lipid bilayer was crucial for accurate alignments. A mask of the lipid bilayer was created by segmenting the best reconstruction with Segger (v1.9.5)[48]. Density subtraction of the lipid bilayer was then performed in RELION followed by masked refinement[49]. This was repeated for a total of two subtractions as residual membrane density was observed. The final map was reconstructed from the original particles with the optimised alignments.

Per frame B-factor weighting was performed by movie refinement and particle polishing of the final subset of particles after classification and refinement of all reconstructions, except in the case of localised reconstructions where polishing was performed prior to sub-particle extraction. Lastly, polished particles were re-refined. For reconstructions below 3 Å, additional corrections and refinements were performed as follows. Ewald sphere correction, astigmatism and beam tilt refinement were performed in RELION (v3.0b2)[50]. In the case of super-resolution images, the data were resampled to the original pixel size and the final iteration of refinement was continued in RELION. Although these manipulations did not yield improvements for most reconstructions, Ewald sphere effects did appear to affect the MPEG1$_{L425K}$α C16 reconstruction, marginally improving the resolution by 0.04 Å.

Global resolution was calculated by the gold standard Fourier shell correlation (FSC) at the 0.143 criterion (Supplementary Fig. 11). Local resolution was estimated for all reconstructions in RELION using a windowed FSC$_{0.143}$ (Supplementary Fig. 12). For B-factor sharpening, MonoRes and LocalDeblur were used to re-estimate local resolution and enhance high resolution features by local sharpening respectively[51]. Locally sharpened maps were subsequently filtered by local-resolution with blocfilt[52] or with RELION. Analysis of resolution anisotropy was performed by 3DFSC[53]. Any conversions between software were performed with EMAN (v2.2), code written in-house or by D. Asarnow and J. Rubinstein.

**Model building and analysis**. Phenix real-space refinement was performed on all models, followed by manual verification of Ramachandran values and fit-to-density within Coot[54]. Analysis of model and map quality was performed by a combination of EMRinger[55] and MolProbity[56] scores. At last, the map-to-model FSC was calculated in Phenix. All figures and visualisation of models, maps and trajectories were performed in UCFS ChimeraX[57], Pymol[58] or VMD[59].

Model building of the MACPF region was originally performed de novo into the MPEG1$_{WT}$ reconstruction in Coot; however, the MABP region was poorly resolved and could not be built due to symmetry breaking. Later, the MABP domain was clearly resolved in both the MPEG1$_{L425K}$β D16 and MPEG1$_{L425K}$α C1 reconstructions (Supplementary Figs. 13–16). The MPEG1$_{L425K}$ α and β conformations differ substantially only in the structure of the L-domain. The L-domain of the α conformation is essentially identical to that of wild-type protein (Supplementary Fig. 17). In contrast, the structure of the β conformation reveals that the mutation has resulted in a domain swapping event between the double ring assembly, such that the two rings are now tightly linked via a β-sandwich formed via interaction of L-domains from opposing molecules (Supplementary Fig. 9d). We were unable to find any evidence of the β conformation in preparations of wild-type protein, accordingly, we suggest that this structure is an artefact induced through the L425K mutation.

These maps were then used to build the remainder of the MABP domain de novo in Coot. The MPEG1$_{L425K}$α C1 model was used as a template to make the final model of MPEG1$_{WT}$, this was fitted into the MPEG1$_{WT}$ reconstruction by rigid body docking followed by Phenix real-space refinement.

A model of the lipid bound MPEG1 prepore was obtained by molecular dynamics flexible fitting. Here, the reconstruction of liposome/MPEG1$_{L425K}$ was used as an energy potential map and a single ring of the MPEG1$_{L425K}$α conformer was flexibly guided into the map by namd2/MDFF[60]. As the L-domain becomes disordered upon lipid binding, this region was removed from the model.

**AFM**. AFM experiments were carried out on a Multimode 8 system operated in peak-force tapping mode with MSNL-E and PFHR-B cantilevers (Bruker, Santa Barbara, USA). In brief, force–distance curves were recorded at frequencies between 2 and 4 kHz with a maximum tip-sample separation of between 5 and 20 nm. Typically, data were collected at a rate of 0.2–1 frame min$^{-1}$. Image analysis was performed on either the open-source SPM analysis software, Gwyddion (v2.53) or the Nanoscope Analysis software (v1.7). Images were plane-levelled, and line-by-line flattened using the lipid membrane as a reference. An additional Gaussian filter was applied with a full-width at half-maximum of two pixels (typical pixel size

of 2 nm) to remove high frequency noise. *E. coli* total lipid extract, 1-palmitoyl-2-oleoyl-glycero-3-phosphocholine (POPC) and 1-palmitoyl-2-oleoyl-sn-glycero-3-phospho-L-serine (POPS) were purchased from Avanti Polar Lipids (Alabaster, USA). Lipids were mixed in chloroform, dried under a stream of nitrogen gas and the resulting film was suspended in buffer (150 mM NaCl, 20 mM HEPES, pH 7.4) to a final concentration of 1 mg ml$^{-1}$. Small unilamellar vesicles with a nominal diameter of 50 nm were produced by extrusion of a multilamellar suspension through a polycarbonate membrane. To obtain an extended, supported lipid bilayer film, 6 μL of the small unilamellar vesicle solution was injected onto a freshly cleaved mica surface (⌀ 9.9 mm, Agar Scientific) covered in 80 μL of adsorption buffer (150 mM NaCl, 20 mM HEPES, 25 mM MgCl$_2$, pH 7.4), and subsequently incubated for 30 min at room temperature. Prior to injecting protein, the supported lipid bilayer was gently washed 15 times with 80 μL of the adsorption buffer to remove residual lipid vesicles. MPEG1 was injected onto the lipid bilayer to a concentration of 70–350 nM and incubated for 15 min at 37 °C. The resulting membrane bound, mobile MPEG1 assemblies were imaged with AFM. To immobilise membrane bound MPEG1, the assemblies were cross-linked by addition of ~ 0.12 % glutaraldehyde (TAAB Laboratories) and incubated for 10 min at room temperature, before imaging with AFM. At high (350 nM) concentrations of MPEG1, the MPEG1 assemblies formed a two-dimensional lattice in the membrane, with sufficient lateral stabilisation to facilitate AFM imaging of the assemblies without glutaraldehyde fixation.

**Reporting summary.** Further information on research design is available in the Nature Research Reporting Summary linked to this article.

## Data availability

Data supporting the findings of this manuscript are available from the corresponding authors upon reasonable request. The source data underlying Fig. 1f, and Supplementary Fig. 8b are provided as a Source Data file. Cryo-EM maps and atomic models have been deposited in the Electron Microscopy Data Bank (EMDB) under accession codes EMBD 20616, EMBD 20617, EMBD 20619, EMBD 20620, EMBD 20621, EMBD 20622, EMBD 20623 and EMBD 20627. Each EMDB entry includes five maps: (1) the low-pass-filtered map without amplitude correction; (2) the low-pass-filtered map with amplitude correction by a negative *B*-factor shown in Extended Data Table 1; (3) the half maps without any post-processing such as low-pass-filtering or amplitude correction; and (4) the mask associated with refinement and sharpening. Coordinates are available from the RCSB Protein Data Bank under accession codes PDB 6U23, PDB 6U2J, PDB 6U2K, PDB 6U2L and PDB 6U2W.

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

## Acknowledgements

J.C.W. is an Australian Research Council (ARC) Laureate Fellow and an Honorary National Health and Medical Research Council (NHMRC) Senior Principal Research Fellow. He acknowledges the previous support of an Australian Research Council (ARC) Federation Fellowship. MAD acknowledges the support of an ARC Future Fellowship and further acknowledges the previous support of an NHMRC Career Development Award. C.B.J. acknowledges the support of the Australian Government RTP scholarship. AWH acknowledges support by the UCL Grand Challenge scheme and the Sackler Foundation. B. W.H. acknowledges support from the UK Biotechnology and Biological Sciences, Medical and Engineering and Physical Sciences Research Councils (BBSRC, BB/N015487/1, MRC, MR/R000328/1 and EPSRC, EP/M028100/1). We thank the staff of the Monash Ramaciotti Centre for Electron Microscopy, the Monash protein production and proteomics platforms, and the support of the MASSIVE supercomputer team. We thank Professor Helen Saibil and Natalya Lukoyanova (Birkbeck College, London) for helpful discussions.

## Author contributions

J.C.W. and M.A.D. conceived the study and co-led the work. J.C.W., M.A.D., S.S.P. and C.B.J. co-wrote the paper. S.S.P. and C.B.J. collected data, performed computational analysis and determined the structures. E.S.P., A.W.H. and B.W.H. performed and analysed AFM experiments. B.W.H. led AFM studies. S.S.P., C.B.J., B.A.S., S.M.E. and P.J.C. produced and analysed protein. H.V., M.R. and G.R. set up collection of EM experiments. P.I.B., I.V., Y.G. and E.S. performed, analysed and co-led extensive experimentation to characterise protein.

## Additional information

**Competing interests:** The authors declare no competing interests.

