## [Peer Review File · Nature Communications]

Reviewers' Comments:

Reviewer #1:

Remarks to the Author:

Pang et al. report on the cryo-EM structure of a hexadecameric assembly of the Macrophage Expressed Gene-1 (MPEG-1), an endosomal/phagosomal perforin-like protein displaying the features of a soluble pre-pore complex. The authors used a spliced version of the MPEG-1 lacking its cytosolic regions and its transmembrane part that natively anchors the protein to the phagolysosomal membrane. Membrane perforation by the pre-pore complexes could be induced by mild acidification as occurring within maturing phagolysosomes. Interestingly, MPEG-1 formed complexes with liposomes via its MABP domain, but unlike conventional perforins, its pore-forming machinery was found to point away from the bound membrane. Thus the authors suggest two possible scenarios how MPEG-1 might function to form pores in engulfed microbes: Proteolytic cleavage of the MPEG-1 ectodomain from the transmembrane domain and/or lowering of pH might permit the release of the pre-pore complex into the phagolysosomal lumen where it may function similar to conventional perforins. Alternatively and most interestingly, MABP domain binding to the phagolysosomal membrane in combination with the pore-forming machinery pointing away from the membrane might provide a mechanism that prevent auto-lysis of the phagolysosome, while simultaneously enable pore formation in engulfed bacteria.

My overall impression of the manuscript is that it has the potential to provide new mechanistic insights in the functioning of phagolysosomes, however, the authors should consider the following suggestions/comments to strengthen the drawn conclusions:

- How did the authors choose the lipid compositions used for the respective experiments: For cryo-EM experiments a 50/50 mixture of POPC/POPS was used. Is such a membrane a valid approximation of the phagolysosomal membrane?
- Why did the authors on the other hand use a different lipid composition (E. coli total lipid extract) in their AFM experiments that they use to confirm MPEG-1 membrane binding? E. coli total lipid extract might be a good model for a bacterial membrane, but likely not a good approximation of a phagolysosomal membrane.
- E.coli lipid extract could be used as a model for a bacterial (target) membrane. Thus a comparison of cryo-EM data or AFM images obtained from MPEG-1 on E.coli lipids and MPEG-1 on a POPC/POPS membrane (as used for the cryo-EM experiments) could provide evidence for a different orientation (as suggested in the manuscript) on the actual target (bacterial membrane) as compared to the phagolysosomal membrane.
- AFM experiments (on a proper membrane) under low pH could be easily used to judge whether pH lowering induces MPEG-1 dissociation from the membrane (Fig. 4a) or not. It could even provide evidence for a pH induced conformational change in case no membrane dissociation is observed (Fig. 4b)

Minor issues:

- It is not clear to the reviewer how pH change alone would permit a release of MPEG-1 into the phagosomal lumen without additional cleavage of the membrane anchor (line 155; and/or): "One possibility is that other intra-cellular events, for example proteolysis⁷ and / or changes in pH (reported here), permits release of MPEG-1 into the lumen of the phagosome" It might rather be that both, proteolytic cleavage and pH reduction have to occur to release MPEG-1 from the membrane. Might there be a physiological reason for this two-step release?
- Height scale in Fig 2d: why starting at -9 nm? There are no features which are that deep.

- Fig. 4a: Proteolysis should not largely change the pre-pore position relative to the membrane (as it is sketched.)

- Which AFM setup was used?

- Generally, Supplementary figure should be referenced in the main article (see guidelines), which is only the case for Suppl. Figures 1-4. Supplementary Figures 5 and 6 are not referenced at all, and Suppl. Figures 7 and 8 only in the methods section.

Reviewer #2:

Remarks to the Author:

This paper presents a structural study of the archetypal MACPF protein, MPEG-1. It reports a primarily structural study of a truncated construct of MPEG-1 that lacks its putative transmembrane and cytosolic domains. The truncated construct retains motifs that are sufficient to facilitate interactions with a cellular membrane and the authors report several structures which ultimately inform two different structural arrangements – a high resolution structure of the soluble “pre-pore” form, assembled as a weakly-associated dimer that seems unlikely to represent a physiologically relevant assembly, as well as a liposome-associated but non-membrane perforating form that is likely to more accurately reflect the physiologically-relevant pre-pore conformation.

The structures are interesting and reveal mostly expected similarities to other pore-forming proteins, including related members of the MACPF family. There are some key differences that are highlighted in the manuscript, including the inverted nature of the pore-forming domain, relative to the membrane surface, when compared to other MACPF-containing proteins. This orientation forms the basis of a hypothesis posited by the authors; that the full length MPEG-1 may remain associated with the phagolysosomal membrane while puncturing a juxtaposed membrane of an engulfed micro-organism, although evidence for this beyond that inferred from the aforementioned structural orientation is not provided in the current study.

The work performed in the manuscript is technically noteworthy. The image processing was clearly challenging, necessitating a number of complex image processing approaches and the resolution of the membrane-associated structure is particularly impressive, given that this was achieved using reconstituted liposomes. Typically, structural resolutions in this range would only be achievable using a synthetic lipid mimetic system such as lipid nanodiscs, although in this regard the authors probably have an ideal system in that the extra-membranous region is quite large and the transmembrane region is either small or non-existent. The structural data are complemented by some activity assays and AFM imaging, but for the most part the focus of the manuscript is on the EM analyses.

While the hypothesis about the mechanism of action associated with MPEG-1 is an obvious one to pose based on the structural data presented, it is most apparent that the manuscript would benefit from the inclusion of additional data that might strengthen this hypothesis. The only evidence given in the current manuscript is the class average proposed to represent a pore structure in Figure 3, which does appear to represent a complex associated with one membrane and perforating a second one, but could equally be an artefact of the crowded liposomal environment. Any supporting evidence (non-structural?) that the authors were able to provide would greatly strengthen the manuscript and ultimately, the likelihood that it may influence thinking in the field.

Notwithstanding the comments above and elaborated below on the scientific content of the manuscript, in my opinion the most important improvements that need to be made concern the way the results are presented and communicated. In order to improve accessibility to the wider scientific community, and also to ensure that there is no ambiguity in statements and claims, there needs to be significant revision of the manuscript structure and writing. Overall the manuscript is

very brief, particularly in the introduction. There are no section headings and detailed explanations are lacking in many areas of the manuscript. This may be because the manuscript has been transferred internally from a different Nature family journal with a more concise format? Regardless of the reason for this, I believe the paper has the potential to be made significantly more accessible to a wider audience by undertaking a significant rewrite and expanding on a number of points, which are flagged in the comments below.

I recommend that the authors consider making major revisions to their manuscript, taking into account the comments below.

Specific points

- The only discussion of the biological background to the study is a statement that MPEG-1 “functions within the macrophage phagolysosome” with no detailed description of how or why. A more comprehensive introduction and summary of the relevant literature will, in my opinion, greatly enhance the accessibility of the manuscript to a readership not familiar with MPEG-1, and may also help to contextualise the later statement that “some patients suffering from pulmonary nontuberculous mycobacterial infections carry MPEG-1 mutations.”
- The chopping and changing between biological role and molecular details in the introductory section (only two short paragraphs), with neither elaborated on particularly well, also impacts readability and while these details may be well established for those familiar with the system, a paper published in Nature Communications should be targeting a wider readership and as such should do a better job in this regard.
- Also in the introduction (line 54), the authors describe MPEG-1 as having three domains, including “a transmembrane domain that spans the vesicular membrane” but then go on to say that “While a bacteriocidal pore-forming function for MPEG-1 has not yet been definitively identified in vivo or in vitro”. The authors should clarify this section of the manuscript so that there is a distinction between the TM domain, which likely anchors MPEG-1 in a membrane, and the MACPF domain, which is independently implicated in establishing pores in lipid membranes.
- Fig S1 – Several terms mentioned in the main text to describe the similarities between MPEG-1, perforin and C9 are not annotated on Fig S1. It would improve the clarity and correspondence between the text and the figure to ensure terms like ectodomain, transmembrane tether etc. are used to identify the relevant regions of the MPEG-1 sequence. Also, rather than the large macrophage cartoon, which is relatively uninformative (and in my opinion unnecessary given that it is also presented in Fig. 4), it would instead be more useful to further annotate this figure in a way that the construct used for structural studies in this work is clearly demarcated and identified.
- Fig 1 – Neither here nor elsewhere in the manuscript do the authors present an image of the model within the EM density in a way that allows the goodness of fit and detail of the map to be subjectively assessed on a wider scale. Two small selected areas are represented in Fig. S7a. I would recommend the authors expand this figure to show a global fitting as well as to show representative areas of well resolved and poorly resolved densities. They should do this for both structures (the soluble dimeric structure and the membrane-associated structure) and for all maps that have been interpreted at the level of atomic models.
- Line 71 the declaration that the two rings “are loosely associated with respect to each other” seems unclear to me. I suspect the authors mean that the two rings are not positioned in a single defined orientation or position, relative to one another, leading them to conclude that they are only loosely associated. This should be clarified as one is an observation and one is a conclusion. The conclusion that this occurs “as a consequence of two membrane interacting surfaces interacting” could be better worded, and it should be clarified whether the double-ringed structures observed for other pore-forming proteins are only observed in vitro or occur in vivo.
- Line 75 “The structure revealed that each MPEG-1 monomer within the 16-subunit assembly comprises an N-terminal MACPF/CDC domain...” – Fig S1A suggests that this could already be predicted based on sequence alone? Was it really the structure that revealed this, or was it merely confirmed? Can the authors re-emphasise what it is that the structure actually reveals that could not have been predicted based on sequence information? In general, the manuscript could do a better job of emphasising the novel aspects of the structure.

- Fig S2 – The workflow presented here suggests that two datasets were collected for the WT MPEG-1. Please clarify if this is the case? If not, why were these datasets processed separately? Were efforts made to combine them?
- Also in Fig S2 – A reconstruction of the point mutant (MPEG-1L425K) is shown, but this mutant is not introduced anywhere in the main text of the manuscript and is only mentioned briefly in the methods -line 210 “A similar approach was used to produce a mutant form of MPEG-1L425K. While the structures yielded by this mutant are not discussed extensively in this paper, the information derived from this structure proved important for determining the complete structure of wild type protein.” This key experimental point should be introduced and acknowledged far earlier in the manuscript. If the reconstruction of the mutant was important to ultimately solving the structure, then I would argue that it is important enough to discuss this in the context of the current manuscript.
- The paragraph starting on line 81 disrupts the discussion of the MPEG-1 structure. A brief comment that MPEG-1 differs from other MACPF/CDCs is already made on line 90 with appropriate citations. I would therefore recommend removing the extended discussion of other systems from this part of the manuscript and instead elaborating on this elsewhere. Perhaps the discussion would be the best place so that the authors can combine this with a discussion of the insights their own structure provides into potential mechanisms of assembly and pore formation.
- Fig 1f – Lytic activity of pre-pore material. Rather than presenting the data as values adjusted for control measurements, can they be presented as sample measurements and control measurements on the same set of axes? Error bars are only visible for the measurement at pH 5. Are the other error bars too small to see? If so this should be noted in the legend. Did the authors try the same pH experiment with a different buffer system to verify that the effect is certainly pH dependent and not dependent on the changing buffer salt?
- An SDS-PAGE gel or some other result that conveys relative purity of the sample should be included for the sample used in the lytic activity assays. The Methods section states that SDS PAGE and SEC were used for purification; including these data should be sufficient.
- Can the authors elaborate on the potential significance of acid activation in an in vivo context?
- Fig 2b: ‘A glycan moiety (indicated by an asterisk) is attached to TMH-2. What is the evidence for this? Is there a known glycosylation site in the corresponding region of the sequence? The resolution seems unlikely to be able to unequivocally trace the structure described by this density?
- According to the methods, POPC/POPS liposomes were used for the MPEG-1 liposome associated cryo-EM structure? This should be mentioned in the results and also discussed in terms of the likely relevance concerning the ‘positively charged hairpin loop’ identified in the MABP domain. Is this lipid mix essential for membrane association? Were other mixtures investigated? If so is there a biological significance that can be discussed here (e.g. role of PS in recognition by macrophages).
- For the AFM studies, E. coli lipids (mostly PE and PG) are used. Is it proposed that the ring-shaped structures assembled on the E. coli lipid membranes represent the pores associated with bacterial membranes via the MACPF domain? Or does this instead indicate that association via the MABP domain is not specific for the POPC/POPS mixture identified above? This point should be discussed. Were any experiments conducted to investigate whether specific lipids are required for membrane association? Such studies may well deliver important biological insights that would greatly enhance the scientific merit of the paper.
- I would recommend including the monomer structure of MPEG-1 in Supp Fig 4 to facilitate comparison with the two structures shown here.
- Paragraph starting line 117 – the statement that the data “reveal that the MABP domain binds lipids in a canonical fashion through the MABP hairpin” is not supported by the data presented. There is no direct evidence of lipid binding presented. Rather, the data can be interpreted as suggesting this, or perhaps consistent with it. Also in this paragraph, I am confused by the statement that membrane interaction leads to a breakage of interactions between the L domain and MABP hairpin – it is difficult to reconcile interactions between the L domain and the hairpin based on the presentation in 2c and furthermore, the two appear to move closer together in the lipid associated form.

- Line 138-139 “The implication of our findings is therefore that MPEG-1 may be able to bind one membrane system through the MABP domain” – if I have understood correctly, aren't the authors suggesting that this first membrane is actually recognised via the coordinated action of the TM domain (absent from their construct) AND the MABP domain?
- The authors mention that extensive efforts were made to obtain a sample that would allow structure determination of a membrane inserted conformation, including incubation of liposome/MPEG-1 at acidic pH (based on the results of the lytic assays). Can the authors provide more details about these experiments? Can the authors speculate as to what might be driving aggregation at acidic pH? Is this liposome aggregation? If so were liposomes reconstituted from different lipid mixtures or charge screening trials investigated (see comments above as well). If different lipids were investigated, what lipids were screened? Alternatively, is liposome stability at acidic pH an issue? If so, does a lipid-stabilising agent such as cholesterol help? Were synthetic membrane mimetics (e.g. nanodiscs, SMALPs, SapNPs) investigated? The authors identified that in rare cases pore forms could be observed in their current datasets. The reasons for this might be worth commenting on in the manuscript. If the problem lies in producing a protein sample that spans two membranes, did the authors try working with further truncated or mutated protein constructs that would be predicted to contact only a single membrane?
- Methods – it is not entirely clear if and at what point symmetry was assumed or determined and then imposed. This should be made clear in the methods. If symmetry was imposed, how was this determined?
- The authors should include model-map FSC plots for those models in Fig S6 that had models built into them. It may be easiest to add these on the same set of axes as the masked and unmasked FSC plots.
- The discussion (concluding two paragraphs of the main text) should be expanded and improved. Given the paucity of data presented by the authors supporting their own conclusions about the functional mechanisms of MPEG-1 (drawn almost entirely from the structure), it would also be appropriate to present a more thorough discussion of how their hypothesis is or isn't supported by other studies in the field.
- Also in the conclusion, there is a statement “Our data further reveals the intriguing finding that at neutral pH the MABP domain binds membranes in an orientation that would preclude MACPF pore formation in the MABP-bound membrane” (line 149-151). If this is indeed a major conclusion of the manuscript, then this was not clear to me prior to this point in the manuscript. An effort should be made to emphasise this finding earlier in the manuscript, and figures prepared that clearly highlight this key finding. I am not disputing that this is the case, just that it is not made clear prior to this point. If the authors are referring to the fact that the MACPF domain is inverted then this paragraph should be re-written to make it clear this is what they mean.
- The final paragraph and the associated Figure 4 warrant a much fuller discussion of potential mechanisms and supporting evidence that they propose for the in vivo activity of MPEG-1. In its current format, this is far too brief.

Other points

- There is some unusual capitalisation in places. For example, the first words of Membrane Attack Complex / Perforin-like / Cholesterol Dependent Cytolysin are capitalised where first defined but Macrophage expressed gene-1 is not. Similarly “cryo-Electron Microscopy”, “Human” etc. are capitalised but shouldn't be. Some abbreviations that have already been defined in the abstract are re-defined later in the manuscript text. The abbreviations should either be removed from the abstract or used in their abbreviated form in first subsequent use.
- The statement that the MABP domain is a fold known to bind lipids needs a reference (line 109).
- Line 117 – is there a word missing here? I'm not sure I understand the phrasing that MPEG-1 pre-pores coordinate membranes? The word “coordinated” is used in several places to describe interaction between MPEG-1 and the membrane, but I don't think this is the right terminology to describe such an interaction or association.
- Line 125 has a double plural (studies...confirms).
- Line 137 is missing a word.
- The legend to Figure 2 is truncated. Figure 2d is incompletely described.

- The components of Figure 3 should be labelled (A, B, C, etc.) for ease of reference.
- Supplementary Figures 5-8 are not cited at all in the main text of the manuscript. Supplementary Figure 3 is an unnecessary inclusion.
- The terminology “pre-pore” and “prepore” are used inconsistently throughout the text.
- The tense regularly changes between past and present in the methodology section (often within the same paragraph).
- Line 313 “However, the quality of the reconstruction in the MABP domain remained uninterpretable due to intramolecular...” I am sure the authors do not mean the quality was uninterpretable – rather that the quality was low and so the density was uninterpretable in this region?
- Line 331 “in in RELION”
- Line 343 Rubinstein is misspelled
- Line 348 - How is model to map quality assessed with molprobit?

Reviewer #3:

Remarks to the Author:

MPEG-1, also known as Perforin 2 is a fascinating protein, conserved across metazoa, that seems to represent an ancient member of perforin-like proteins. It is proposed to function within the phagolysosome to damage engulfed microbes. MPEG-1 is a transmembrane protein that bears a lysosomal targeting sequence in the cytoplasmic tail and lacks the C2 domains of perforin that allow membrane binding. The current manuscript provides a structure for MPEG-1 which provides important new insights that will greatly facilitate the understanding of its biological role. This analysis provides a number of interesting new insights that lead to the proposal of a model in which proteolysis or pH change releases MPEG-1 into the lumen of the phagolysosome where it can form pores. The structure also suggests an alternative model in which MPEG1 could bind both the phagolysosome membrane and simultaneously form pores in engulfed bacteria.

The current manuscript is well written, but I think that the paper would be more accessible to the reader if the authors could clarify a few points, and make a few modifications to the presentation of the figures as outlined: -

line 88 appears to have a typo where complete appears as compete.

line 111 is missing the reference for “previous studies reveal MABP proteins interact with acidic membranes via a positively charged loop.”

Supp Fig 4 is supposed to show that the MABP domain is inverted compared to the C2 membrane binding domains of perforin. This is difficult to appreciate when only structures for perforin and LLO are given without any labels on the domains and without MPEG-1 in the same figure. Flipping back and forth with Figures 1c and 2a is confusing as the color schemes are different for equivalent structural domains between each. Please make sure that domain colors are consistent throughout the paper and add the MPEG-1 structure to Supp Fig4 along with labels for the domains in each structure. This will make the paper much easier for readers to follow.

Our responses to the comment or question are in blue, bold font.

Actual changes to the manuscript are shown in red, bold font.

Reviewers' comments:

Reviewer #1 (Remarks to the Author):

1. How did the authors choose the lipid compositions used for the respective experiments: For cryo-EM experiments a 50/50 mixture of POPC/POPS was used. Is such a membrane a valid approximation of the phagolysosomal membrane?

The choice of lipid composition was guided by lipid binding assays using lipid strips and SPR (results not included). Both assays suggested that MPEG can bind to a range of lipids but with slightly higher affinities for acidic lipids, such as POPS. Therefore POPS was included in the liposomes to give a higher proportion of lipid-bound MPEG-1 for cryo-EM analysis. In this particular study, the POPC/POPS (1:1 mix) liposomes produced the best grids for data cryo-EM collection, however, we wish to avoid any inference that this is the optimal lipid membrane. Indeed, in these regards we note that the precise composition of the phagolysosomal membrane remains controversial. We also observed that MPEG-1 can bind *in vitro* in a similar orientation to liposomes generated from bacterial membranes (Supp. Fig. 5). These data suggest that the MABP domain may be relatively non-selective with respect to lipid composition.

2. Why did the authors on the other hand use a different lipid composition (E. coli total lipid extract) in their AFM experiments that they use to confirm MPEG-1 membrane binding? E. coli total lipid extract might be a good model for a bacterial membrane, but likely not a good approximation of a phagolysosomal membrane.

The AFM studies were not performed with the intention of confirming the cryo-EM experiments, rather the AFM experiments were performed in parallel. We chose a bacterial (i.e., potential target) model membrane for the AFM experiments because we were aiming to observe membrane-inserted (and hence more static, easier to resolve) MPEG-1 assemblies upon lowering of the pH. At the time of the experiment, we were unaware of the orientation of the MABP domain and the possibility that there were two independent membrane interacting surfaces, i.e. the side with the MABP domains and side with the pore .

Subsequent EM experiments on liposomes derived from bacterial membranes now reveal that the MPEG-1 MABP domain can interact *in vitro* with bacterial membranes in a similar fashion to POPC/POPS membranes (Supp. Fig. 5). These data suggest that the MABP domain may be relatively non-selective with respect to lipid composition, and also explain why we are unable to resolve static pores upon dropping the pH (see point 4/5 below).

Manuscript change: We have altered the phrasing in the manuscript to reflect this: "Separate atomic force microscopy (AFM) studies, performed concurrently with EM studies, also confirm the binding of MPEG1 prepores to membranes (Fig. 2d)."

3. E.coli lipid extract could be used as a model for a bacterial (target) membrane. Thus a comparison of cryo-EM data or AFM images obtained from MPEG-1 on E.coli lipids and MPEG-1 on a POPC/POPS membrane (as used for the cryo-EM experiments) could provide evidence for a different orientation (as suggested in the manuscript) on the actual target (bacterial membrane) as compared to the phagolysosomal membrane.

We have performed this experiment and included these EM data as requested (Supp. Fig 5b). These experiments reveal that the MPEG-1 MABP domain can interact *in vitro* with bacterial membranes in a similar fashion to POPC/POPS membranes (Supp. Fig. 5a). This result suggests that the MABP domain is relatively non selective in terms of lipid composition (observed in membrane lipid strips) and that pathogen targeting *per se* is unlikely to be achieved solely through lipid binding specificity. Taken together and considering the trafficking and overall structure of MPEG-1, we suggest that MABP domain is initially driven to interact with phagolysosomal membrane through the position of the type I transmembrane domain. In this model, MPEG-1 would thus be "pre-bound" via the MABP domain to the host membrane prior to encountering bacterial membranes (Fig. 4).

4. AFM experiments (on a proper membrane) under low pH could be easily used to judge whether pH lowering induces MPEG-1 dissociation from the membrane (Fig. 4a) or not.

and

5. It could even provide evidence for a pH induced conformational change in case no membrane dissociation is observed (Fig. 4b)

On the bacterial model membranes (E. coli lipids), we did attempt to lower the pH and found large disruption to the membrane leading to generation of debris to the extent that we could not obtain meaningful AFM images any more. We have yet to determine the conditions under which the pH treatment can occur in a more controlled, gradual way. With respect to performing these experiments on a phagolysosomal model membrane, we expect to encounter similar imaging difficulties due to debris and aggregation. Possibly due to the *in trans* pore state (c.f. Fig. 3) whereby hydrophobic regions become exposed leading to aggregation.

No manuscript changes made at this stage.

Minor issues:

6. It is not clear to the reviewer how pH change alone would permit a release of MPEG-1 into the phagosomal lumen without additional cleavage of the membrane anchor (line 155; and/or): “One possibility is that other intra-cellular events, for example proteolysis and / or changes in pH (reported here), permits release of MPEG-1 into the lumen of the phagosome” It might rather be that both, proteolytic cleavage and pH reduction have to occur to release MPEG-1 from the membrane. Might there be a physiological reason for this two-step release?

This sentence was changed in the manuscript to read “One possibility is that other intra-cellular events, for example proteolysis prior to pH change, permits release of a portion of MPEG1 into the lumen of the phagosome”

7. Height scale in Fig 2d: why starting at -9 nm? There are no features which are that deep.

The darker (more negative) part of the colour scale has now been removed.

8. Fig. 4a: Proteolysis should not largely change the pre-pore position relative to the membrane (as it is sketched.)

The figure labels have been amended to make the illustration clearer, here we are depicting diffusion away from the membrane after the possible proteolytic event(s).

9. Which AFM setup was used?

The following sentence has been added to the manuscript “AFM experiments were carried out on a Multimode 8 system with MSNL-E cantilevers (Bruker, Santa Barbara, USA) using parameters, setups and analysis as previously described”

10. Generally, Supplementary figure should be referenced in the main article (see guidelines), which is only the case for Suppl. Figures 1-4. Supplementary Figures 5 and 6 are not referenced at all, and Suppl. Figures 7 and 8 only in the methods section.

This has been rectified in the manuscript.

Reviewer #2 (Remarks to the Author):

1. The only discussion of the **biological background** to the study is a statement that MPEG-1 “functions within the macrophage phagolysosome” with no detailed description of how or why. A **more comprehensive introduction and summary of the relevant literature** will, in my opinion, greatly enhance the accessibility of the manuscript to a readership

not familiar with MPEG-1, and may also help to contextualise the later statement that “some patients suffering from pulmonary nontuberculous mycobacterial infections carry MPEG-1 mutations.”

The introduction has been expanded to include important background information for MPEG-1 and MACPF/CDC.

2. The **chopping and changing between biological role and molecular details in the introductory section** (only two short paragraphs), with neither elaborated on particularly well, also impacts readability and while these details may be well established for those familiar with the system, a paper published in Nature Communications should be targeting a wider readership and as such should do a better job in this regard.

We have addressed the flow and readability of the introduction.

3. Also in the introduction (line 54), the authors describe MPEG-1 as having three domains, including “a transmembrane domain that spans the vesicular membrane” but then go on to say that “While a bacteriocidal pore-forming function for MPEG-1 has not yet been definitively identified in vivo or in vitro”. The authors should clarify this section of the manuscript so that there is a **distinction between the TM domain, which likely anchors MPEG-1 in a membrane, and the MACPF domain, which is independently implicated in establishing pores** in lipid membranes.

The phrasing has been adjusted in the manuscript to further clarify the distinction between the roles of the transmembrane region and the MACPF domain and the established pore forming function

- **“human MPEG-1 is an integral type-1 transmembrane protein”**
- **“a transmembrane domain anchor that spans the vesicular membrane”**
- **“MPEG-1 is proposed to function within the macrophage phagolysosome as a pore-forming protein via the MACPF/CDC domain”**

4. Fig S1 – Several **terms mentioned in the main text to describe the similarities between MPEG-1, perforin and C9 are not annotated on Fig S1**. It would improve the clarity and correspondence between the text and the figure to ensure terms like ectodomain, transmembrane tether etc. are used to identify the relevant regions of the MPEG-1 sequence. Also, rather than the large macrophage cartoon, which is relatively uninformative (and in my opinion unnecessary given that it is also presented in Fig. 4), it would instead be more useful to further annotate this figure in a way that the construct used for structural studies in this work is clearly demarcated and identified.

This has been addressed in an amended version of the figures, included as Supplementary Figure 2.

5. Fig 1 – Neither here nor elsewhere in the manuscript do the authors present an image of the model within the EM density in a way that allows the goodness of fit and detail of the map to be subjectively assessed on a wider scale. Two small selected areas are represented in Fig. S7a. I would recommend the authors expand this figure to show a global fitting as well as to show representative areas of well resolved and poorly resolved densities. They should do this for both structures (the soluble dimeric structure and the membrane-associated structure) and for all maps that have been interpreted at the level of atomic models.

This has been addressed as Supplementary Figures 11-14.

6. Line 71 the declaration that the two rings **“are loosely associated with respect to each other”** seems unclear to me. I suspect the authors mean that the two rings are not positioned in a single defined orientation or position, relative to one another, leading them to conclude that they are only loosely associated. This should be clarified as one is an observation and one is a conclusion. The conclusion that this occurs **“as a consequence of two membrane interacting surfaces interacting”** could be better worded, and it should be clarified whether the double-ringed structures observed for other pore-forming proteins are only observed in vitro or occur in vivo.

Analysis by multi-body refinement, as well as 2D and 3D classification, revealed that the two rings have several degrees of freedom relative to one another and indeed are not in a single well-defined orientation or position.

We postulate that this interaction is not physiological, rather an effect of our recombinant expression construct *in vitro*. The association appears to be driven by the presence of two exposed MABP-membrane binding interfaces which have a propensity for charged surfaces due to the MABP β -hairpin. When MPEG-1 is bound to a membrane, this interface would be relatively shielded from solvent and be in close proximity to the charged phospholipid headgroups. In the absence of a membrane, the propensity of the MABP-membrane interacting interface for charged surfaces (such as a membrane) may be responsible for driving the non-specific and loose association of the two rings.

We have added a small discussion in the results section of the manuscript that clarifies this point about the range of orientation of the two rings relative to each other. “Three-dimensional classification experiments of the head-to-head assembly reveal that one ring can adopt multiple positions relative to the other ring, that is, neither ring possess a single well-defined orientation. (Supp. Fig. 2). Similar “double ring” structures have been observed in other pore forming proteins (e.g. aerolysin; gasdermin) and can form as a consequence of two membrane binding surfaces interacting with each other. Indeed, in these regards the MPEG-1 interaction is mediated by the interface defined by the MABP. This is likely to be an *in vitro* effect of our recombinant construct, which lacks the transmembrane region (Supp. Fig. 2).”

The line “as a consequence of two membrane interacting surfaces interacting” has been changed to “as a consequence of two membrane binding surfaces interacting with each other”.

7. Line 75 “The structure revealed that each MPEG-1 monomer within the 16-subunit assembly comprises an N-terminal MACPF/CDC domain...” – Fig S1A suggests that this could already be predicted based on sequence alone? Was it really the structure that revealed this, or was it merely confirmed? Can the authors re-emphasise what it is that the structure actually reveals that could not have been predicted based on sequence information? In general, the manuscript could do a better job of emphasising the novel aspects of the structure.

We have rewritten the manuscript to reflect this point.

The paragraph has been amended to “The structure confirmed previous bioinformatic prediction of the MACPF/CDC domain. In contrast, the MPEG1 MABP domain possesses <5% primary amino acid sequence similarity to other homologues and the fold was accordingly identified using Dali searches”.

8. Fig S2 – The workflow presented here suggests that two datasets were collected for the WT MPEG-1. Please clarify if this is the case?

Only a single dataset was collected. Two separate, initial particle populations were processed independently, as a retrospective analysis had to be performed in light of results with the L425K mutation and with the release of new software.

If not, why were these datasets processed separately?

Retrospective analysis was performed after solving the L425K structure as insights were gained into the specimen behaviour. With the advent of new software, a separate analysis was performed altogether.

Were efforts made to combine them?

Not applicable as there is only one dataset.

No manuscript changes.

9. Also in Fig S2 – A reconstruction of the point mutant (MPEG-1L425K) is shown, but this mutant is not introduced anywhere in the main text of the manuscript and is only mentioned briefly in the methods -line 210 “A similar approach was used to produce a mutant form of MPEG-1L425K. While the structures yielded by this mutant are not discussed extensively in this paper, the information derived from this structure proved important for determining the complete structure of wild

type protein.” This key experimental point should be introduced and acknowledged far earlier in the manuscript. If the reconstruction of the mutant was important to ultimately solving the structure, then I would argue that it is important enough to discuss this in the context of the current manuscript.

We have modified the sentence in the first results paragraph starting “Accordingly, an integrative single particle cryo-EM approach was used to elucidate the structure of this assembly. These data revealed that MPEG1 formed hexadameric rings stacked together to form a “double-ring” (or head-to-head assembly). While this structure was incomplete, particularly in the C-terminal region, these data provided a platform for directed mutagenesis. Indeed, our (ultimately unsuccessful) attempts to disrupt the interface between the double ring serendipitously led to our identifying the MPEG1 L425K mutation as yielding a greatly improved sample and an improvement in global resolution to 2.4 Å (the workflow described in detail in the methods). This mutation resulted in two different MPEG1_{L425K} assemblies, one of which involved domain swapping between the two head-to-head coordinated rings. Ultimately, a cohort of structures were determined, modelled and interpreted.”

For completeness we have also included a brief description of the L425K structure (β conformation) in the methods.

10. The paragraph starting on line 81 disrupts the discussion of the MPEG-1 structure. A brief comment that MPEG-1 differs from other MACPF/CDCs is already made on line 90 with appropriate citations. I would therefore recommend removing the extended discussion of other systems from this part of the manuscript and instead elaborating on this elsewhere. Perhaps the discussion would be the best place so that the authors can combine this with a discussion of the insights their own structure provides into potential mechanisms of assembly and pore formation.

We have moved the paragraph discussing the general mechanism into the introduction.

11. Fig 1f – Lytic activity of pre-pore material. Rather than presenting the data as values adjusted for control measurements, can they be presented as sample measurements and control measurements on the same set of axes?

Examples of raw data have been included in Supplementary Figure 6. Since the protein response is logarithmic, in order to easily visualise a turbidity response at various pH the protein concentration is adjusted for clarity.

Error bars are only visible for the measurement at pH 5. Are the other error bars too small to see? If so this should be noted in the legend.

Yes, the error bars become too small to see on the other measurements. We have updated the figure legend to clarify.

Did the authors try the same pH experiment with a different buffer system to verify that the effect is certainly pH dependent and not dependent on the changing buffer salt?

Different buffer systems were tested during assay optimisation for example glycine, MES, HEPES, sodium cacodylate buffers. These experiments gave similar results as to those reported in the article. We could not vary the ionic conditions of the buffer (kept close to isotonic conditions) because these affected the structural integrity of red blood cells leading to unwanted lysis.

12. An SDS-PAGE gel or some other result that conveys relative purity of the sample should be included for the sample used in the lytic activity assays. The Methods section states that SDS PAGE and SEC were used for purification; including these data should be sufficient.

An SDS-PAGE example has been included as Supplementary Figure 6a.

13. Can the authors elaborate on the potential significance of acid activation in an in vivo context?

Previously MPEG-1 has been shown to localise throughout the endosomal pathway. Ubiquitination was recently shown to also play a regulatory role in the redistribution and trafficking of MPEG-1. After detection of LPS or other proinflammatory immune cytokines, MPEG-1 becomes ubiquitinated and trafficks to the late endosome and phagolysosome. These compartments typically contain a low pH lumen. If this process is blocked, i.e. by mutation of ubiquitination sites, MPEG-1 trafficking is stalled and this results in a null phenotype.

In these regards, our data further supports a role of acid in MPEG-1 regulation. We observe that an increase in MPEG-1 activity is correlated with a decrease in pH. Our data and the literature are therefore consistent with a role in targeting intracellular pathogens after phagocytosis in acidic compartments. This model would also suggest an additional level of negative regulation that prevents MPEG-1 mediated autolysis, or premature pore formation, prior to localisation to lysosome/phagosome.

A paragraph has been added that expands upon the potential significance of low pH upon MPEG-1 activation. "Furthermore, the subcellular localisation of MPEG1 within the endosomal pathway, is consistent with MPEG1 encountering pathogens in a low pH environment."

14. Fig 2b: 'A glycan moiety (indicated by an asterisk) is attached to TMH-2. What is the evidence for this?'

We observe additional electron density on asparagine residues flanked by canonical N-glycosylation motifs. The N-glycans on residues N168 and N252 are very well resolved and have been modelled in some reconstructions. This has been included in supplementary figures 11-14. Due to the excellent resolution, we felt that we did not need to confirm the N-glycans using mass spectrometry.

Is there a known glycosylation site in the corresponding region of the sequence?

To the authors knowledge, no studies have confirmed the presence of these post-translational modifications. Four N-glycosylation sites are predicted based on analysis of the primary structure. Two of these have been modelled into density, however density for the other two is not observed.

The resolution seems unlikely to be able to unequivocally trace the structure described by this density?

There is clear and unequivocal density for the first two mannose moieties of N-glycans located at in the C1 (2.8 Å) and D16 (2.8 Å) reconstructions. This has been included in the supplementary figures 11-14.

15. According to the methods, POPC/POPS liposomes were used for the MPEG-1 liposome associated cryo-EM structure?

Yes, this is correct.

This should be mentioned in the results and also discussed in terms of the likely relevance concerning the 'positively charged hairpin loop' identified in the MABP domain.

This is now detailed.

Is this lipid mix essential for membrane association?

This particular mixture of POPC and POPS is not essential. Membrane association occurs on a range of lipids. Lipid binding experiments using SPR and membrane lipid strips data (not presented here) suggests MPEG-1 binds to a wide range of lipids, but has a higher affinity (approximately 10 fold) for phospholipids with acidic head groups. This behaviour is also observed for ESCRT MABP (Boura and Hurley, 2012). The POPS/POPC mixture was empirically chosen to maximise the number of membrane associated complexes in order to improve the likelihood of a successful data collection.

Were other mixtures investigated?

Yes, to study affinity and specificity. Studies were also performed by electron microscopy, however, the best sample for data collection was obtained by use of the described lipid composition. See also response to reviewer 1.

If so is there a biological significance that can be discussed here (e.g. role of PS in recognition by macrophages).

To the best of our knowledge, the lipid composition of the endosomal pathway is not very well established. The inner leaflet of the endosomal pathway is reported to be enriched with phospholipids such as POPS and this would be consistent with adhering MPEG-1 to the inner leaflet of the endosomal vesicles during trafficking. A similar hypothesis is suggested to occur for the trafficking of the previously studied ESCRT complex (Boura and Hurley, 2012).

16. For the AFM studies, E. coli lipids (mostly PE and PG) are used.

Is it proposed that the ring-shaped structures assembled on the E. coli lipid membranes represent the pores associated with bacterial membranes via the MACPF domain?

See response to reviewer 1, comments #2 and #3.

Or does this instead indicate that association via the MABP domain is not specific for the POPC/POPS mixture identified above? This point should be discussed.

See response to reviewer 1. SPR and membrane lipid strip analysis suggests the MABP association has broad specificity for negatively charged phospholipids - including PE found in E. coli membranes. Our data suggest the MABP β -hairpin alone is not sufficient to distinguish between host and pathogen membranes. We propose that the transmembrane helix may initially orient the MABP β -hairpin toward the host membrane, thereby directing the MACPF domain away from the host vesicular membrane into the lumen of the vesicle (Fig. 4).

Were any experiments conducted to investigate whether specific lipids are required for membrane association? Such studies may well deliver important biological insights that would greatly enhance the scientific merit of the paper.

SPR and membrane lipid strips were used to test various lipid compositions (not shown). We found MPEG-1 shows broad specificity to lipids that typically have negative charge. Similar results were reported for the ESCRT MABP domain by Boura & Hurley (2012).

17. I would recommend including the monomer structure of MPEG-1 in Supp Fig 4 to facilitate comparison with the two structures shown here.

We agree and have amended the figure accordingly.

We have amended Supp. Fig. 4 to include an MPEG-1 monomer, as well as domain labels.

18. Paragraph starting line 117 – the statement that the data “reveal that the MABP domain binds lipids in a canonical fashion through the MABP hairpin” is not supported by the data presented. There is no direct evidence of lipid binding presented.

Rather, the data can be interpreted as suggesting this, or perhaps consistent with it. Also in this paragraph, I am confused by the statement that membrane interaction leads to a breakage of interactions between the L domain and MABP hairpin – it is difficult to reconcile interactions between the L domain and the hairpin based on the presentation in 2c and furthermore, the two appear to move closer together in the lipid associated form.

The authors agree that the original figure fails to clearly illustrate this point and we have amended it.

To clarify the original statement, the statement now reads “the latter region bends as a consequence of interactions with the membrane, a shift that involves breaking the in trans contacts with the L-domain of the adjacent subunit”.

19. Line 138-139 “The implication of our findings is therefore that MPEG-1 may be able to bind one membrane system through the MABP domain” – if I have understood correctly, aren’t the authors suggesting that this first membrane is actually recognised via the coordinated action of the TM domain (absent from their construct) AND the MABP domain?

Yes the reviewer has correctly understood, our data suggests both the MABP and TM domains may act redundantly with respect to one another. The precise maturation pathway of MPEG-1 is, however, unknown and is not the focus of this study.

20. The authors mention that extensive efforts were made to obtain a sample that would allow structure determination of a membrane inserted conformation, including incubation of liposome/MPEG-1 at acidic pH (based on the results of the lytic assays).

Can the authors provide more details about these experiments?

Can the authors speculate as to what might be driving aggregation at acidic pH?

We suggest that acid activation leads to exposure of additional hydrophobic, MACPF membrane-penetrating regions of MPEG-1 that may cause substantial aggregation as well as liposome rupture.

Is this liposome aggregation?

The liposomes no longer seem intact. The protein also appears to clump presumably via the hydrophobic membrane spanning regions of the MPEG-1 MACPF domain.

If so were liposomes reconstituted from different lipid mixtures or charge screening trials investigated (see comments above as well).

Yes. See response to reviewer #1, comments #2, #3.

If different lipids were investigated, what lipids were screened?

Yes. See response to reviewer #1, comments #2, #3.

Alternatively, is liposome stability at acidic pH an issue?

No, controls were always included to ensure that liposome stability was maintained under acidic conditions (in the absence of MPEG-1).

If so, does a lipid-stabilising agent such as cholesterol help?

Cholesterol was included in some of the experiments and the authors observed no difference.

Were synthetic membrane mimetics (e.g. nanodiscs, SMALPs, SapNPs) investigated?

Yes. We have tried several membrane mimetics. Including amphipols, SMALPs, saposinA, detergents and nanodiscs to solubilise MPEG-1 pores. None have been successful so far.

The authors identified that in rare cases pore forms could be observed in their current datasets. The reasons for this might be worth commenting on in the manuscript. If the problem lies in producing a protein sample that spans two membranes,

did the authors try working with further truncated or mutated protein constructs that would be predicted to contact only a single membrane?

Yes, the authors have tried many additional constructs (various truncations, many point mutations). None have been successful so far.

21. Methods – it is not entirely clear if and at what point symmetry was assumed or determined and then imposed. This should be made clear in the methods. If symmetry was imposed, how was this determined?

The following sentences have been added in the manuscript. “Symmetry was determined from 2D and 3D class averages and correct symmetry was confirmed by comparison to C1 reconstructions. Ultimately different symmetries were imposed depending on the reconstruction ranging from full D16, C16 or C1.”

22. The authors should include model-map FSC plots for those models in Fig S6 that had models built into them. It may be easiest to add these on the same set of axes as the masked and unmasked FSC plots.

The authors agree and these have been included in Supp. Fig. 9.

23. The discussion (concluding two paragraphs of the main text) **should be expanded and improved**. Given the paucity of data presented by the authors supporting their own conclusions about the functional mechanisms of MPEG-1 (drawn almost entirely from the structure), it would also be appropriate to present a more thorough discussion of how their hypothesis is or isn't supported by other studies in the field.

We have expanded the discussion as requested.

24. Also in the conclusion, there is a statement “Our data further reveals the intriguing finding that at neutral pH the MABP domain binds membranes in an orientation that would preclude MACPF pore formation in the MABP-bound membrane” (line 149-151). If this is indeed a major conclusion of the manuscript, then this was not clear to me prior to this point in the manuscript. An effort should be made to emphasise this finding earlier in the manuscript, and figures prepared that clearly highlight this key finding. **I am not disputing that this is the case, just that it is not made clear prior to this point.** If the authors are referring to the fact that the MACPF domain is inverted then this paragraph should be re-written to make it clear this is what they mean.

We have clarified this point throughout the manuscript.

25. The final paragraph and the associated Figure 4 warrant a much fuller discussion of potential mechanisms and supporting evidence that they propose for the in vivo activity of MPEG-1. In its current format, this is far too brief.

We have expanded the discussion, in balance with avoiding excessive speculation.

Other points

26. There is some unusual capitalisation in places. For example, the first words of Membrane Attack Complex / Perforin-like / Cholesterol Dependent Cytolysin are capitalised where first defined but Macrophage expressed gene-1 is not. Similarly “cryo-Electron Microscopy”, “Human” etc. are capitalised but shouldn't be. Some abbreviations that have already been defined in the abstract are re-defined later in the manuscript text. The abbreviations should either be removed from the abstract or used in their abbreviated form in first subsequent use.

Incorrect use of capitals has been addressed in the manuscript.

27. The statement that the MABP domain is a fold known to bind lipids needs a reference (line 109).

The appropriate citation has been included.

28. Line 117 – is there a word missing here? I’m not sure I understand the phrasing that MPEG-1 pre-pores coordinate membranes? The word “coordinated” is used in several places to describe interaction between MPEG-1 and the membrane, but I don’t think this is the right terminology to describe such an interaction or association.

We use the word coordinate to avoid excessive repetition of the word bound or bind.

29. Line 125 has a double plural (studies...confirms).

The word “confirms” has been changed to “confirm”.

30. Line 137 is missing a word

Addressed.

31. The legend to Figure 2 is truncated. Figure 2d is incompletely described.

The figure legend has been amended.

32. The components of Figure 3 should be labelled (A, B, C, etc.) for ease of reference.

The figure has been amended.

33. Supplementary Figures 5-8 are not cited at all in the main text of the manuscript. Supplementary Figure 3 is an unnecessary inclusion.

All supplementary figures are now referred to in the main text.

34. The terminology “pre-pore” and “prepore” are used inconsistently throughout the text.

All instances of pre-pore have been changed to prepore.

35. The tense regularly changes between past and present in the methodology section (often within the same paragraph).

The appropriate tenses have now been used.

36. Line 313 “However, the quality of the reconstruction in the MABP domain remained uninterpretable due to intramolecular...” I am sure the authors do not mean the quality was uninterpretable – rather that the quality was low and so the density was uninterpretable in this region?

This has been rephrased “However, due to intramolecular, conformational heterogeneity between subunits the quality of the reconstruction in the MABP domain remained poor and hence the electron density was uninterpretable.”

37. Line 331 “in in RELION”

The typo has been amended.

38. Line 343 Rubinstein is misspelled.

The typo has been amended.

39. Line 348 - How is model to map quality assessed with molprobity?

This has been rephrased to “Analysis of map and model quality was performed by a combination of EMRinger (Adams et al., 2015) and MolProbity (Chen et al., 2010) scores respectively.”

Reviewer #3 (Remarks to the Author):

1. line 88 appears to have a typo where complete appears as compete.

We have corrected this typo.

2. line 111 is missing the reference for “previous studies reveal MABP proteins interact with acidic membranes via a positively charged loop.”

The appropriate reference has been cited (Boura and Hurley, 2012)

3. Supp Fig 4 is supposed to show that the MABP domain is inverted compared to the C2 membrane binding domains of perforin. This is difficult to appreciate when only structures for perforin and LLO are given without any labels on the domains and without MPEG-1 in the same figure.

A monomer of MPEG-1 has been included in this figure for comparison and labels have been added.

4. Flipping back and forth with Figures 1c and 2a is confusing as the color schemes are different for equivalent structural domains between each. Please make sure that domain colors are consistent throughout the paper and add the MPEG-1 structure to Supp Fig4 along with labels for the domains in each structure. This will make the paper much easier for readers to follow.

We have verified that the colour scheme is consistent and where possible labels have been included for clarity.

Reviewers' Comments:

Reviewer #1:

Remarks to the Author:

The authors have satisfyingly addressed all my previous comments and suggestions. They have further expanded the introductory paragraph and improved the readability of the manuscript. I can now recommend publication without any reservation.

Minor issues:

- Reference style does not seem to be Nature style
- The same is true for the figure captions

Reviewer #2:

Remarks to the Author:

The manuscript by Pang et al. is vastly improved from the original submission. Substantial attention has been given to redrafting the manuscript and it now reads much better.

I have 3 main comments about the revised manuscript.

1. Supplementary Tables: No data collection and refinement statistics have been made available. These are referred to in the table legend of Table S2, but I could not find the table itself anywhere? In addition, the colour coding in Table S1 is not explained or defined anywhere that I could see?

2. Fig S3 - an explanation of the two separate processing pipelines for the initial MPEG-1 WT data is provided in the rebuttal letter; this explanation should be included in the manuscript as well (e.g. in the Methods section) to avoid readers encountering the same confusion. Also, several different but specific aspects of the processing workflow are identified at various points in the manuscript (e.g. the reference to the 3D classification experiments on line 145). To assist with directing the reader, it may be useful to label pertinent parts of the figure for ease of reference? Better still, an enlarged version of the 3D classification panel could even be included as an additional figure so that the mentioned structural heterogeneity is more readily visualised as the complexity of this figure necessitates that it can only be quite small as a panel in Fig S3.

3. In their rebuttal letter, the authors refer to what appears to be a fairly extensive series of experiments that investigate the lipid dependence of MPEG1 membrane affinity. I am curious as to why the authors chose not to include this data in their manuscript as it would support some of the claims made in the paper with regard to physiological context. Specifically, the data would support the authors assertions that MPEG1 does not have a specific lipid affinity, and that this implicates mechanisms similar to those defined previously for endosomal sorting complexes.

In my opinion, the manuscript would be much stronger if these data were incorporated. Given that Reviewer 1 also raised this question during review I feel it is an important point to address in the manuscript, rather than in dialogue with the reviewers or as data not shown. Inclusion of these data would also help to strengthen the potential impact and significance of the manuscript, which in its current form still relies very heavily on conclusions derived entirely from structural analysis and largely untested hypotheses.

Reviewers' comments:

Reviewer #1 (Remarks to the Author):

The authors have satisfyingly addressed all my previous comments and suggestions. They have further expanded the introductory paragraph and improved the readability of the manuscript. I can now recommend publication without any reservation.

Minor issues:

- Reference style does not seem to be Nature style
- The same is true for the figure captions

These points have been addressed.

Reviewer #2 (Remarks to the Author):

The manuscript by Pang et al. is vastly improved from the original submission. Substantial attention has been given to redrafting the manuscript and it now reads much better.

I have 3 main comments about the revised manuscript.

1. Supplementary Tables: No data collection and refinement statistics have been made available. These are referred to in the table legend of Table S2, but I could not find the table itself anywhere? In addition, the colour coding in Table S1 is not explained or defined anywhere that I could see?

Supplementary table 2 was uploaded at the time of submission as an additional file as per the nature guidelines, we have verified this document is included again at the time of resubmission.

Colour coding has now been defined in the figure caption as the following, "The contacts found around the central lumen of the β -barrel are shaded in red, and the *in trans* subunit interactions between the L-domain and the adjacent MABP β -hairpin are shaded in green."

2. Fig S3 - an explanation of the two separate processing pipelines for the initial MPEG-1 WT data is provided in the rebuttal letter; this explanation should be included in the manuscript as well (e.g. in the Methods section) to avoid readers encountering the same confusion.

The following sentence has been included in the methods section, "Resultantly, a similar analysis was performed on the MPEG1_{WT} data set also yielding an improvement from 3.5 to 2.9 Å (although the advent of newer software from RELION-1.4 to 3.0 may also explain this improvement)."

Also, several different but specific aspects of the processing workflow are identified at various points in the manuscript (e.g. the reference to the 3D classification experiments on line 145). To assist with directing the reader, it may be useful to label pertinent parts of the figure for ease of reference? Better still, an enlarged version of the 3D classification panel could even be included as an additional figure so that the mentioned structural heterogeneity is more readily visualised as the complexity of this figure necessitates that it can only be quite small as a panel in Fig S3.

We have further clarified Fig S3 by including the means of particle coordinate identification at the beginning of each workflow.

To further guide the reader, we have clarified in the main text that a combination of 3D classification and multibody analysis provided insights into the structural heterogeneity of the D16 assembly. We have also included a new supplementary figure highlighting 2D classes that illustrate this conformational variation (Fig S5).

3. In their rebuttal letter, the authors refer to what appears to be a fairly extensive series of experiments that investigate the lipid dependence of MPEG1 membrane affinity. I am curious as to why the authors chose not to include this data in their manuscript as it would support some of the claims made in the paper with regard to physiological context. Specifically, the data would support the authors assertions that MPEG1 does not have a specific lipid affinity, and that this

implicates mechanisms similar to those defined previously for endosomal sorting complexes.

The following paragraph has been modified to include the experiments performed to identify lipid specificity, “Given these findings we sought to understand MPEG1 lipid specificity and how MPEG1 prepores coordinate membranes. Initial lipid screening by membrane lipid strips suggested that MPEG1, like the MVB12 MABP domain, displays a preference for negatively charged lipids such as POPS, PIPs and cardiolipin (Fig. 2b). Negatively charged lipids were hence empirically chosen to yield data suitable for cryoEM. We next determined the 3.6 Å cryo-EM structure of the MPEG1 prepore assembly bound to POPC/POPS liposomes at neutral pH (Fig. 2c). These data revealed the MPEG1 MABP binds lipids in a canonical fashion through the MABP β -hairpin.”

The following paragraph has been included in the methods section, “Lipid binding screen: The Membrane Lipid Strips were purchased from Echelon Biosciences. The lipid binding screen was carried out as recommended by the supplier with some modifications. The strips were blocked with 1 % (w/v) skim milk in TBS (10 mM Tris-HCl, pH 8.0, 150 mM NaCl) overnight at 4 °C. The blocked strips were incubated with 10 $\mu\text{g mL}^{-1}$ MPEG1 in TBS-T (TBS and 0.1 % (v/v) Tween 20) at room temperature for 1 hour. The strips were washed three times in TBS-T before probing with anti-6 \times His tag-HRP conjugated antibody (Abcam) to detect lipid-protein interactions.”

In my opinion, the manuscript would be much stronger if these data were incorporated. Given that Reviewer 1 also raised this question during review I feel it is an important point to address in the manuscript, rather than in dialogue with the reviewers or as data not shown. Inclusion of these data would also help to strengthen the potential impact and significance of the manuscript, which in its current form still relies very heavily on conclusions derived entirely from structural analysis and largely untested hypotheses.

As noted above, these data on lipid specificity have now been included in the manuscript. In addition, we have included more extensive AFM data on both the POPC/POPS (Suppl. Fig. 7) and *E. coli* lipid extract membranes (Fig. 3), showing no significant differences in MPEG1 prepore formation on these two types of lipids.